# SAFE ONLINE BID OPTIMIZATION WITH RETURN ON INVESTMENT AND BUDGET CONSTRAINTS

## ABSTRACT

In online marketing, the advertisers' goal is balancing between achieving *high volumes* and *high profitability*. The companies business units address this tradeoff by maximizing the volumes while guaranteeing a minimum Return On Investment (ROI) level. Technically speaking, such a task can be naturally modeled as a combinatorial optimization problem subject to ROI and budget constraints that can be solved online. In this picture, the learner's uncertainty over the constraints' parameters plays a crucial role since the algorithms' exploration choices might lead to their violation during the entire learning process. Such violations represent a major obstacle to adopting online techniques in real-world applications. Thus, controlling the algorithms' exploration during learning is paramount to making humans trust online learning tools. This paper studies the nature of both optimization and learning problems. In particular, we show that the learning problem is inapproximable within any factor (unless $P = NP$) and provide a pseudo-polynomial-time algorithm to solve its discretized version. Subsequently, we prove that no online learning algorithm can violate the (ROI or budget) constraints a sublinear number of times during the learning process while guaranteeing a sublinear regret. We provide the $GCB$ algorithm that guarantees sublinear regret at the cost of a linear number of constraint violations, and $GCB_{safe}$ that guarantees w.h.p. a constant upper bound on the number of constraints violations at the cost of a linear regret. Moreover, we designed $GCB_{safe}(\psi, \phi)$, which guarantees both sublinear regret and safety w.h.p. at the cost of accepting tolerances $\psi$ and $\phi$ in the satisfaction of the ROI and budget constraints, respectively. Finally, we provide experimental results to compare the regret and constraint violations of $GCB$ and $GCB_{safe}$.

## 1 INTRODUCTION

Nowadays, Internet advertising is *de facto* the leading advertising medium. Notably, while the expenditure on physical ads, radio, and television has been stable for a decade, that on Internet advertising is increasing with a ratio of $\approx 20\%$ per year, reaching the considerable amount of 189 billion USD in 2021 only in the US IAB (2021). Internet advertising has two main advantages over traditional channels: it provides precise ad targeting and allows an accurate, real-time evaluation of investment performance. On the other hand, the amount of data the platforms provide and the plethora of parameters to be set make its optimization impractical without using artificial intelligence.

The advertisers' goal is usually to set *bids* in the attempt to balance the tradeoff between achieving *high volumes*, corresponding to maximizing the sales of the products to advertise, and *high profitability*, corresponding to maximizing Return On Investment (ROI). The companies' business units need simple ways to address this tradeoff, and customarily, they maximize the volumes while constraining ROI to be above a given threshold. The importance of ROI constraints, in addition to standard budget constraints (i.e., spent per day for advertising), is remarked in several empirical studies. We mention, *e.g.*, the data analysis on the auctions performed on Google's AdX by Golrezaei et al. (2021), showing that many advertisers take into account ROI constraints, particularly in hotel booking. However, no platform provides features to force the satisfaction of ROI constraints, and some platforms (*e.g.*, TripAdvisor, and Trivago) do not even allow the setting of daily budget constraints. Thus, the problem of satisfying those constraints is a challenge that advertisers must address by designing suitable bidding strategies. In this picture, uncertainty plays a crucial role as the revenue and cost of the advertising campaigns are unknown beforehand and need to be estimated online by learning

algorithms during the sequential arrival of data. As a result, the constraints are subject to uncertainty, and wrong parameter estimations can make the ROI and budget constraints arbitrarily violated when using an uncontrolled exploration like that used by classical online learning algorithms. Such violations represent today the major obstacles to adopting AI tools in real-world applications, as advertisers often are unwilling to accept such risks. Remarkably, this issue is particularly crucial in the early stages of the learning process as adopting algorithms with an uncontrolled exploration when a small amount of data is available can make the advertising campaigns' performance oscillate by a large magnitude, which advertisers negatively perceive. Therefore, to make humans trust online artificial intelligence algorithms, controlling their exploration accurately to mitigate risks and provide safety guarantees during the entire learning process is paramount.[1]

**Original Contributions** As customary in the online advertising literature, see, *e.g.*, Devanur & Kakade (2009), we assume stochastic (*i.e.*, non-adversarial) clicks, and we adopt Gaussian Processes (GPs) to model the problem. Let us remark that the assumption that clicks are generated stochastically is reasonable in practice as the advertising platforms can limit manipulation due to malicious bidders. For instance, Google Ads can identify invalid clicks and exclude them from the advertisers' spending. In this paper, we study the nature of both the optimization and learning problems. Initially, we focus on studying our optimization problem without uncertainty, showing that no approximation within any strictly positive factor is possible with ROI and budget constraints unless $\mathsf{P} = \mathsf{NP}$, even in simple, realistic instances. However, when dealing with a discretized space of the bids as it happens in practice, the problem admits an exact pseudo-polynomial time algorithm based on dynamic programming. Most importantly, when the problem is with uncertainty, we show that no online learning algorithm can violate the ROI and/or budget constraints a sublinear number of times while guaranteeing a sublinear pseudo-regret. We provide an online learning algorithm, namely $\mathsf{GCB}$, providing pseudo-regret sublinear in the time horizon $T$ at the cost of a linear number of violations of the constraints. We also provide its safe version, namely $\mathsf{GCB_{safe}}$, guaranteeing w.h.p. a constant upper bound on the number of constraints' violations at the cost of a regret linear in $T$. Inspired by the two previous algorithms, we design a new algorithm, namely $\mathsf{GCB_{safe}}(\psi, \phi)$, guaranteeing both the violation w.h.p. of the constraints for a constant number of times and a pseudo-regret sublinear both in $T$ and the maximum information gain of the GP when accepting tolerances $\psi$ and $\phi$ in the satisfaction of the ROI and budget constraints, respectively. Finally, we performed an empirical study of our algorithms in terms of pseudo-regret/number of constraint violations tradeoff in simulated settings, showing the importance of adopting safety constraints and the effectiveness of our algorithms.

## 2 OPTIMIZATION PROBLEM

We are given an advertising campaign $\mathcal{C} = \{C_1, \dots, C_N\}$, with $N \in \mathbb{N}$ and where $C_j$ is the $j$-th subcampaign and a finite time horizon of $T \in \mathbb{N}$ rounds (each corresponding to one day in our application). In this work, as common in the literature on ad allocation optimization, we refer to a subcampaign as a single ad or a group of homogeneous ads requiring setting the same bid. For every round $t \in \{1, \dots, T\}$ and every subcampaign $C_j$, an advertiser needs to specify the bid $x_{j,t} \in X_j$, where $X_j \subset \mathbb{R}^+$ is a finite set of values that can be set for subcampaign $C_j$. For every round $t \in \{1, \dots, T\}$, the goal is to find the values of bids maximizing the overall cumulative expected revenue while keeping the ROI above a fixed value $\lambda \in \mathbb{R}^+$ and the budget below a daily value $\beta \in \mathbb{R}^+$. Formally, the resulting constrained optimization problem at round $t$ is as follows:

$$\max_{(x_{1,t}, \dots, x_{N,t}) \in X_1 \times \dots \times X_N} \sum_{j=1}^{N} v_j \, n_j(x_{j,t}) \tag{1a}$$

$$\text{s.t.} \quad \frac{\sum_{j=1}^{N} v_j \, n_j(x_{j,t})}{\sum_{j=1}^{N} c_j(x_{j,t})} \geq \lambda, \tag{1b}$$

$$\sum_{j=1}^{N} c_j(x_{j,t}) \leq \beta, \tag{1c}$$

where $n_j(x_{j,t})$ and $c_j(x_{j,t})$ are the expected number of clicks and the expected cost given the bid $x_{j,t}$ for subcampaign $C_j$, respectively, and $v_j$ is the value per click for subcampaign $C_j$. We remark

---

[1]A discussion on the literature related to the topic of advertising with ROI and budget constraints is deferred to Appendix A due to space reasons.

that Constraint (1b) is the ROI constraint, forcing the revenue to be at least $\lambda$ times the costs, and Constraint (1c) keeps the daily spend under a predefined overall budget $\beta$.

At first, we show that, even if all the values of the parameters of the optimization problem are known, the optimal solution cannot be approximated in polynomial time within any strictly positive factor (even depending on the size of the instance) unless $\mathsf{P} = \mathsf{NP}$.

**Theorem 1** (Inapproximability). *For any $\rho \in (0, 1]$, there is no polynomial-time algorithm returning a $\rho$-approximation to the problem in Equations (1a)-(1c), unless $\mathsf{P} = \mathsf{NP}$.*

The proof follows from a reduction of our problem from SUBSET-SUM that is an NP-hard problem.[2,3] It is well known that SUBSET-SUM is a weakly NP-hard problem, admitting an exact algorithm whose running time is polynomial in the size of the problem and the magnitude of the data involved rather than the base-two logarithm of their magnitude. The same can be shown for our problem. Indeed, we can design a pseudo-polynomial-time algorithm to find the optimal solution in polynomial time w.r.t. the number of possible values of revenues and costs. In real-world settings, the values of revenue and cost are in limited ranges and rounded to the nearest cent, allowing the problem to be solved in a reasonable time. Therefore, in what follows, we assume to be given a discretization of the daily costs and revenue value ranges.

## 3 ONLINE LEARNING PROBLEM FORMULATION

We focus on the case in which $n_j(\cdot)$ and $c_j(\cdot)$ in Equations (1a)-(1c) are unknown functions whose values need to be estimated online. This goal is achieved using their noisy realizations $\tilde{n}_{j,h}(x_{j,t})$ and $\tilde{c}_{j,h}(x_{j,t})$ obtained setting the bid $x_{j,t}$ on the $j$-th subcampaign at time $h$. Our problem can be naturally modeled as a multi-armed bandit where the available *arms* are the different values of the bid $x_{j,t} \in X_j$ satisfying the combinatorial constraints of the optimization problem.[4] More specifically, our goal is to design a *learning policy* $\mathfrak{U}$ returning, at every round $t$, a *super-arm* $\{\hat{x}_{j,t}\}_{j=1}^N$, i.e., an arm profile specifying one bid per subcampaign.[5] Since the policy $\mathfrak{U}$ can only use estimates of the unknown number-of-click and cost functions built using past realizations $\tilde{n}_{j,h}(x_{j,t})$ and $\tilde{c}_{j,h}(x_{j,t})$ with $h < t$, the solutions returned by policy $\mathfrak{U}$ may not be optimal and/or violate Constraints (1b) and (1c) when evaluated with the true values. Therefore, we must design $\mathfrak{U}$ so that the violations occur only for a limited number of rounds over the time horizon $T$.

We are interested in evaluating learning policies $\mathfrak{U}$ in terms of revenue losses (a.k.a. pseudo-regret) and safety regarding ROI and budget constraints violations.

**Definition 1** (Learning policy pseudo-regret). *Given a learning policy $\mathfrak{U}$, the* pseudo-regret *is:*

$$R_T(\mathfrak{U}) := T\, G^* - \mathbb{E}\left[\sum_{t=1}^T \sum_{j=1}^N v_j\, n_j(\hat{x}_{j,t})\right],$$

*where $G^* := \sum_{j=1}^N v_j\, n_j(x_j^*)$ is the expected revenue provided by a clairvoyant algorithm, the set of bids $\{x_j^*\}_{j=1}^N$ is the optimal clairvoyant solution to the problem in Equations (1a)-(1c), and the expectation $\mathbb{E}[\cdot]$ is taken w.r.t. the stochasticity of the learning policy $\mathfrak{U}$.*

**Definition 2** ($\eta$-safe learning policy). *Given $\eta \in (0, T]$, a learning policy $\mathfrak{U}$ generating an allocation $\{\hat{x}_{j,t}\}_{j=1}^N$ is $\eta$-safe if the expected number of times it violates at least one of the Constraints (1b) and (1c) from $t = 1$ to $t = T$ is less than $\eta$ or, formally:*

$$\sum_{t=1}^T \mathbb{P}\left(\frac{\sum_{j=1}^N v_j\, n_j(\hat{x}_{j,t})}{\sum_{j=1}^N c_j(\hat{x}_{j,t})} < \lambda \vee \sum_{j=1}^N c_j(\hat{x}_{j,t}) > \beta\right) \le \eta.$$

---

[2]Given a set $S$ of integers $u_i \in \mathbb{N}^+$ and an integer $z \in \mathbb{N}^+$, SUBSET-SUM requires to decide whether there is a set $S^* \subseteq S$ with $\sum_{i \in S^*} u_i = z$.

[3]For space reasons, the proofs are deferred to the Supplementary Material.

[4]Here, we assume that the value per click $v_j$ is known. Refer to Nuara et al. (2018) if it is unknown.

[5]Even if this setting is closely related to the one studied by Badanidiyuru et al. (2013), the non-matroidal nature of the constraints does not allow to cast the bid allocation problem above into the bandit-with-knapsack framework.

---

**Algorithm 1** GCB and GCB$_{\texttt{safe}}$ pseudo-code

---

    **Input**: sets of bid values $X_1, \ldots, X_N$, ROI threshold $\lambda$, daily budget $\beta$

1: Initialize the GPs for the number of clicks and costs
2: **for** $t \in \{1, \ldots, T\}$ **do**
3:     **for** $j \in \{1, \ldots, N\}$ **do**
4:         **for** $x \in X_j$ **do**
5:             Compute $\hat{n}_{j,t-1}(x)$ and $\hat{\sigma}^n_{j,t-1}(x)$ according to Eq. equation 2 and equation 3, respectively
6:             Compute $\hat{c}_{j,t-1}(x)$ and $\hat{\sigma}^c_{j,t-1}(x)$ according to Eq. equation 4 and equation 5, respectively
7:     Compute $\boldsymbol{\mu}$ according to Eq. equation 6 or equation 7
8:     Call the optimization subroutine $\mathsf{Opt}(\boldsymbol{\mu}, \lambda)$ to get a solution $\{\hat{x}_{j,t}\}_{j=1}^N$
9:     Set the prescribed allocation $\{\hat{x}_{j,t}\}_{j=1}^N$ during round $t$
10:     Get revenue $\sum_{j=1}^N v_j \, \tilde{n}_j(\hat{x}_{j,t})$ and pay costs $\sum_{j=1}^N \tilde{c}_j(\hat{x}_{j,t})$
11:     Update the GPs using the new information $\tilde{n}_{j,t}(\hat{x}_{j,t})$ and $\tilde{c}_{j,t}(\hat{x}_{j,t})$

---

In principle, we would design *no-regret* algorithms, i.e., whose pseudo-regret $R_T(\mathfrak{U})$ increases sub-linearly w.r.t. $T$, and, at the same time, that are $\eta$-safe, with $\eta$ sublinearly increasing in (or is independent of) $T$. However, the following theorem shows that no online learning algorithm can provide a sublinear pseudo-regret while guaranteeing safety.

**Theorem 2** (Pseudo-regret/safety tradeoff). *For every $\epsilon > 0$ and time horizon $T$, there is no algorithm with pseudo-regret smaller than $(1/2 - \epsilon) T$ and that violates (in expectation) the constraints less than $(1/2 - \epsilon) T$ times.*

This impossibility result is crucial in practice, showing that no online learning algorithm can theoretically guarantee both a sublinear regret and a sublinear number of violations of the constraints. Therefore, advertisers must accept a tradeoff between the two requirements in real-world applications.

## 4 PROPOSED ALGORITHMS

Even if the above-defined problem is closely related to the Combinatorial MAB formulation provided by Chen et al. (2013), designing algorithms for this setting requires dealing with an additional challenge. Indeed, in the classical Combinatorial MAB framework, the set of arms can be chosen from a fixed and known set of arms. Instead, in the framework we defined, the set of constraints influences the set of arms that are feasible according to the constraints. In the following, we propose two algorithms that carefully define the feasible set of arms at each round to deal with this issue.

We provide the pseudo-code of our algorithms, namely GCB and GCB$_{\texttt{safe}}$, in Algorithm 1, which solves the problem in Equations (1a)-(1c) online while guaranteeing sublinear regret or $\eta$-safety, respectively. Algorithm 1 is divided into an *estimation phase* (Lines 3-7) based on Gaussian Processes (GPs) (Rasmussen & Williams, 2006) to model the parameters whose values are unknown, and an *optimization subroutine* to solve the optimization problem once given the estimates (Line 8). Finally, in the last phase, the newly acquired data are used to improve the GP estimates that will be used in the following round (Lines 10-11).

**Estimation Phase** In Algorithm 1, GPs are used to model functions $n_j(\cdot)$ and $c_j(\cdot)$, describing the expected number of clicks and the costs, respectively. The employment of GPs to model these functions provides several advantages w.r.t. other regression techniques, such as a probability distribution over the possible values of the functions for every bid value $x \in X_j$ relying on a finite set of samples. GPs use the noisy realization of the actual number of clicks $\tilde{n}_{j,h}(\hat{x}_{j,h})$ collected from each subcampaign $C_j$ for every previous round $h \in \{1, \ldots, t-1\}$ to generate, for every bid $x \in X_j$, the estimates for the expected value $\hat{n}_{j,t-1}(x)$ and the standard deviation of the number of clicks $\hat{\sigma}^n_{j,t-1}(x)$. Analogously, using the noisy realizations of the actual cost $\tilde{c}_{j,h}(\hat{x}_{j,h})$, with $h \in \{1, \ldots, t-1\}$, GPs generate, for every bid $x \in X_j$, the estimates for the expected value $\hat{c}_{j,t-1}(x)$

and the standard deviation of the cost $\hat{\sigma}_{j,t-1}^c(x)$. Formally, the above values are computed as follows:

$$\hat{n}_{j,t-1}(x) := \boldsymbol{k}_{j,t-1}(x)^\top [K_{j,t-1} + \sigma^2 I]^{-1} \boldsymbol{k}_{j,t-1}(x), \tag{2}$$

$$\hat{\sigma}_{j,t-1}^n(x) := k_j(x,x) - \boldsymbol{k}_{j,t-1}^\top(x)[K_{j,t-1} + \sigma^2 I]^{-1} \boldsymbol{k}_{j,t-1}(x), \tag{3}$$

$$\hat{c}_{j,t-1}(x) := \boldsymbol{h}_{j,t-1}^\top(x)[H_{j,t-1} + \sigma^2 I]^{-1} \boldsymbol{h}_{j,t-1}(x), \tag{4}$$

$$\hat{\sigma}_{j,t-1}^c(x) := h_j(x,x) - \boldsymbol{h}_{j,t-1}^\top(x)[H_{j,t-1} + \sigma^2 I]^{-1} \boldsymbol{h}_{j,t-1}(x), \tag{5}$$

where $k_j(\cdot,\cdot)$ and $h_j(\cdot,\cdot)$ are the kernels for the GPs over the number of clicks and costs, respectively, $K_{j,t-1}$ and $H_{j,t-1}$ are the Gram matrix over the bids selected during the rounds $\{1, \ldots t-1\}$ for the two GPs, $\sigma^2$ is the variance of the noise of the GPs, $\boldsymbol{k}_{j,t-1}(x)$ and $\boldsymbol{h}_{j,t-1}$ are vectors built computing the kernel between the training bids and the current bid $x$, and $I$ is the identity matrix of order $t-1$. For further details on using GPs, we point an interested reader to Rasmussen & Williams (2006).

The estimation subroutine returns the vector $\boldsymbol{\mu}$ of parameters characterizing the specific instance of the optimization problem. More specifically, it is composed as follows:

$$\boldsymbol{\mu} := (\bar{\mathbf{w}}_1, \ldots, \bar{\mathbf{w}}_N, \underline{\mathbf{w}}_1, \ldots, \underline{\mathbf{w}}_N, -\bar{\mathbf{c}}_1, \ldots, -\bar{\mathbf{c}}_N),$$

where $\bar{\mathbf{w}}_j = (\bar{w}_j(x_1), \ldots, \bar{w}_j(x_{|X_j|}))$ and $\underline{\mathbf{w}}_j = (\underline{w}_j(x_1), \ldots, \underline{w}_j(x_{|X_j|}))$, respectively, $\overline{w}_j(x_j) := v_j\,\overline{n}_j(x_j)$ and $\underline{w}_j(x_j) := v_j\,\underline{n}_j(x_j)$ denote different estimates for the revenue of a subcampaign $C_j$, and $\bar{\mathbf{c}}_j = (\bar{c}_j(x_1), \ldots, \bar{c}_j(x_{|X_j|}))$ is the vector characterizing the costs for subcampaign $C_j$. In the optimization subroutine, $\overline{w}_j(x_j)$ and $\underline{w}_j(x_j)$ will be used to compute the value of Equation (1a) and Equation (1b), respectively, and $\bar{c}_j(x_j)$ to compute the value of Equations (1b)-(1c). We use $\overline{h}$ and $\underline{h}$ to denote potentially different estimated values of a generic function $h$ used by the learning algorithms in the next sections. The proposed algorithms are derived from two procedures to compute $\boldsymbol{\mu}$. Formally, we have that the elements of $\boldsymbol{\mu}$ are defined as follows:

$$\text{GCB}: \begin{cases} \overline{w}_j(x) = \underline{w}_j(x) := v_j \left[ \hat{n}_{j,t-1}(x) + \sqrt{b_{t-1}} \hat{\sigma}_{j,t-1}^n(x) \right], \\ \overline{c}_j(x) := \hat{c}_{j,t-1}(x) - \sqrt{b_{t-1}} \hat{\sigma}_{j,t-1}^c(x), \end{cases} \tag{6}$$

$$\text{GCB}_{\texttt{safe}}: \begin{cases} \overline{w}_j(x) := v_j \left[ \hat{n}_{j,t-1}(x) + \sqrt{b_{t-1}} \hat{\sigma}_{j,t-1}^n(x) \right], \\ \underline{w}_j(x) := v_j \left[ \hat{n}_{j,t-1}(x) - \sqrt{b_{t-1}} \hat{\sigma}_{j,t-1}^n(x) \right], \\ \overline{c}_j(x) := \hat{c}_{j,t-1}(x) + \sqrt{b_{t-1}} \hat{\sigma}_{j,t-1}^c(x). \end{cases} \tag{7}$$

where $b_t$ is an uncertainty term that is appropriately set in Section 5.

The GCB and GCB$_{\texttt{safe}}$ algorithm relies on the idea that the elements in the $\boldsymbol{\mu}$ vector represent the statistical upper/lower bounds to the expected values of the number of clicks and costs. This follows a common choice in the bandit literature to incentivize exploration for uncertain quantities (a.k.a. optimism in the face of uncertainty principle).

We remark that Accabi et al. (2018) propose the GCB algorithm to face general combinatorial bandit problems where the arms are partitioned in subsets and the payoffs of the arms belonging to the same subset are modeled with a GP. To obtain theoretical sublinear guarantees on the regret for our online learning problem, we use a specific definition of $\boldsymbol{\mu}$ vector, making Algorithm 1 be an extension of GCB when the payoffs and constraints are functions whose parameters are modeled by multiple independent GPs. With a slight abuse of terminology, we refer to this extension as GCB.

**Optimization Subroutine** The pseudo-code of the $\mathsf{Opt}(\boldsymbol{\mu}, \lambda)$ subroutine, solving the problem in Equations (1a)-(1c) with a dynamic programming approach, is provided in Algorithm 2. It takes as input the set of the possible bid values $X_j$ for each subcampaign $C_j$, the set of the possible cumulative cost values $Y$ such that $\max_{y \in Y} y = \beta$, the set of the possible revenue values $R$, an ROI threshold $\lambda$, and a vector $\boldsymbol{\mu}$ characterizing the optimization problem. In particular, if the functions are known beforehand, it holds $\overline{h} = \underline{h} = h$ for both $h = w_j$ and $h = c_j$. For the sake of clarity, $\overline{w}_j(x)$ is used in the objective function, while $\underline{w}_j(x)$ and $\overline{c}_j(x)$ are used in the constraints. At first, the subroutine initializes a matrix $M$ in which it stores the optimal solution for each combination of values $y \in Y$ and $r \in R$, and it initializes the vectors $\mathbf{x}^{y,r} = \mathbf{x}_{\text{next}}^{y,r} = [\ ], \forall\, y \in Y, \forall\, r \in R$ (Lines 1 and 2, respectively). Then, the subroutine generates the set $S(y,r)$ of the bids for subcampaign $C_1$ (Line 3). More precisely, the set $S(y,r)$ contains only the bids $x$ that induce the overall costs to be lower than or equal to $y$ and the overall revenue to be higher than or equal to $r$. The bid in $S(y,r)$ that maximizes the revenue calculated with parameters $\overline{w}_j$ is included in the vector $\mathbf{x}^{y,r}$, while the corresponding revenue is stored in the matrix $M$ (Lines 4–5). Then, the subroutine iterates over each subcampaign $C_j$, with $j \in \{2, \ldots, N\}$, all the values $y \in Y$, and all the values $r \in R$ (Lines 9–11). At each

---

**Algorithm 2** $\mathsf{Opt}(\boldsymbol{\mu}, \lambda)$ subroutine

---

**Input**: sets of bid values $X_1, \ldots, X_N$, set of cumulative cost values $Y$, set of revenue values $R$, vector $\boldsymbol{\mu}$, ROI threshold $\lambda$

1: Initialize $M$ empty matrix with dimension $|Y| \times |R|$
2: Initialize $\mathbf{x}^{y,r} = \mathbf{x}^{y,r}_{\text{next}} = [\ ], \forall y \in Y, r \in R$
3: $S(y,r) = \bigcup \left\{ x \in X_1 \mid \overline{c}_1(x) \leq y \land \underline{w}_1(x) \geq r \right\} \ \forall y \in Y, r \in R$
4: $\mathbf{x}^{y,r} = \arg \max_{x \in S(y,r)} \overline{w}_1(x) \ \ \forall y \in Y, r \in R$
5: $M(y,r) = \max_{x \in S(y,r)} \overline{w}_1(x) \ \ \forall y \in Y, r \in R$
6: **for** $j \in \{2, \ldots, N\}$ **do**
7: $\quad$ **for** $y \in Y$ **do**
8: $\quad\quad$ **for** $r \in R$ **do**
9: $\quad\quad\quad$ Update $S(y,r)$ according to Equation equation 8
10: $\quad\quad\quad$ $\mathbf{x}^{y,r}_{\text{next}} = \arg \max_{\mathbf{s} \in S(y,r)} \sum_{i=1}^{j} \overline{w}_i(s_i)$
11: $\quad\quad\quad$ $M(y,r) = \max_{\mathbf{s} \in S(y,r)} \sum_{i=1}^{j} \overline{w}_i(s_i)$
12: $\quad$ $\mathbf{x}^{y,r} = \mathbf{x}^{y,r}_{\text{next}}$
13: Choose $(y^*, r^*)$ according to Equation (9)
14: **Output: $\mathbf{x}^{y^*, r^*}$**

---

iteration, for every pair $(y, r)$, the subroutine stores in $\mathbf{x}^{y,r}$ the optimal set of bids for subcampaigns $C_1, \ldots, C_j$ that maximizes the objective function and stores the corresponding optimum value in $M(y,r)$. At every $j$-th iteration, the computation of the optimal bids is performed by evaluating a set of candidate solutions $S(y,r)$, computed as follows:

$$\bigcup \left\{ \mathbf{s} = [\mathbf{x}^{y',r'}, x] \text{ s.t. } y' + \overline{c}_j(x) \leq y \ \land r' + \underline{w}_j(x) \geq r \land x \in X_j \land y' \in Y \land r' \in R \right\}. \quad (8)$$

This set is built by combining the optimal bids $\mathbf{x}^{y',r'}$ computed at the $(j-1)$-th iteration with one of the bids $x \in X_j$ available for the $j$-th subcampaign, such that these combinations satisfy the ROI and budget constraints. Then, the subroutine assigns the element of $S(y,r)$ that maximizes the revenue to $\mathbf{x}^{y,r}_{\text{next}}$ and the corresponding revenue to $M(y,r)$. At the end, the subroutine computes the optimal pair $(y^*, r^*)$ as follows:

$$(y^*, r^*) = \left\{ y \in Y, r \in R \text{ s.t. } \frac{r}{y} \geq \lambda \ \land M(y,r) \geq M(y', r'), \quad \forall y' \in Y, \forall r' \in R \right\}, \quad (9)$$

and the corresponding set of bids $\mathbf{x}^{y^*, r^*}$, one bid per subcampaign.[6] The following property holds:

**Theorem 3** (Optimality). *The $\mathsf{Opt}(\boldsymbol{\mu}, \lambda)$ subroutine returns the optimal solution to the problem in Equations (1a)-(1c) when $\overline{w}_j(x) = \underline{w}_j(x) = v_j\, n_j(x)$ and $\overline{c}_j(x) = c_j(x)$ for each $j \in \{1, \ldots, N\}$ and the values of revenues and costs are in $R$ and $Y$, respectively.*

## 5 Theoretical Guarantees

In what follows, we provide the theoretical guarantees of the $\mathsf{GCB}$ and $\mathsf{GCB}_{\texttt{safe}}$ algorithms in terms of pseudo-regret and $\eta$-safety. Moreover, we will show how to have sublinear guarantee on both of them by allowing small violations of the constraints. Let us first define the *maximum information gain* $\gamma_{j,t}$ of the GP modeling the number of clicks of subcampaign $C_j$ at round $t$, formally defined as:

$$\gamma_{j,t} := \frac{1}{2} \max_{(x_{j,1}, \ldots, x_{j,t}), x_{j,h} \in X_j} \left| I_t + \frac{\Phi(x_{j,1}, \ldots, x_{j,t})}{\sigma^2} \right|,$$

where $I_t$ is the identity matrix of order $t$, $\Phi(x_{j,1}, \ldots, x_{j,t})$ is the Gram matrix of the GP computed on the vector $(x_{j,1}, \ldots, x_{j,t})$, and $\sigma \in \mathbb{R}^+$ is the noise standard deviation.

### 5.1 Guaranteeing Sublinear Pseudo-regret: GCB

The $\mathsf{GCB}$ algorithm, which is based on the optimist in the face of uncertainty principle, provides the following pseudo-regret bound:

---

[6] An analysis of the running time of Algorithm 1 is provided in Appendix C.

**Theorem 4** (GCB pesudo-regret). *Given $\delta \in (0, 1)$, GCB applied to the problem in Equations (1a)-(1c), with probability at least $1 - \delta$, suffers from a pseudo-regret of:*

$$R_T(\textsf{GCB}) \leq \sqrt{\frac{8T v_{\max}^2 N^3 b_T}{\ln(1 + \sigma^2)} \sum_{j=1}^{N} \gamma_{j,T}} \ ,$$

*where $b_t := 2\ln\left(\frac{\pi^2 NQTt^2}{3\delta}\right)$ is an uncertainty term used to guarantee the confidence level required by GCB, $v_{\max} := \max_{j \in \{1,\dots,N\}} v_j$ is the maximum value per click over all subcampaigns, and $Q := \max_{j \in \{1,\dots,N\}} |X_j|$ is the number of bids in a subcampaign.*

We remark that the upper bound provided in the above theorem is expressed in terms of the maximum information gain $\gamma_{j,T}$ of the GPs over the number of clicks. The problem of bounding $\gamma_{j,T}$ for a generic GP has already been addressed by Srinivas et al. (2010), where the authors present the bounds for the squared exponential kernel $\gamma_{j,T} = \mathcal{O}((\ln T)^2)$ for 1-dimensional GPs. Notice that, thanks to the previous result, the GCB algorithm using squared exponential kernels suffers from a sublinear pseudo-regret since the terms $\gamma_{j,T}$ is bounded by $\mathcal{O}((\ln T)^2)$, and the bound in Theorems 4 is $\mathcal{O}(N^{3/2}(\ln T)^{5/2})\sqrt{T}$. However, the GCB algorithm violates (in expectation) the constraints a linear number of times in $T$.

**Theorem 5** (GCB safety). *Given $\delta \in (0, 1)$, GCB applied to the problem in Equations (1a)-(1c) with $b_t = 2\ln\left(\frac{\pi^2 NQTt^2}{3\delta}\right)$ is $\eta$-safe where $\eta \geq T - \frac{\delta}{2NQT}$ and, therefore, the number of constraints violations is linear in $T$.*

This result states that if we apply the GCB algorithm, we expect to have a large revenue over the time horizon $T$ at the cost of violating the ROI and/or the budget constraints most of the time over the learning period. Therefore, in practical cases, such an algorithm might perform poorly regarding ROI over the entire time horizon $T$. As highlighted before, this behaviour might lead to a premature stop of the algorithm from the business unit. In what follows, we overcome this issue by being more conservative in estimating the constraint satisfaction.

### 5.2 GUARANTEEING SAFETY: GCB$_{\texttt{safe}}$

The GCB$_{\texttt{safe}}$ algorithm uses different bounds than GCB to evaluate the constraints and have stronger guarantees about their satisfaction. In particular, while the estimates for the revenue of the algorithm (Equation (1a)) are estimated using upper bounds, for the constraints (Equation (1b)-(1c)), we used statistical lower bounds to guarantee they are satisfied at every round with high probability. This choice comes at the cost of a linear worst-case performance in terms of pseudo-regret:

**Theorem 6** (GCB$_{\texttt{safe}}$ pseudo-regret). *Given $\delta \in (0, 1)$, GCB$_{\texttt{safe}}$ with $b_t := 2\ln\left(\frac{\pi^2 NQTt^2}{3\delta}\right)$, applied to the problem in Equations (1a)-(1c) suffers from a pseudo-regret $R_t(\textsf{GCB}_{\texttt{safe}}) = \Theta(T)$.*

However, GCB$_{\texttt{safe}}$ violates the ROI and budget constraints only a constant number of times w.r.t. $T$.

**Theorem 7** (GCB$_{\texttt{safe}}$ safety). *Given $\delta \in (0, 1)$, GCB$_{\texttt{safe}}$ with $b_t := 2\ln\left(\frac{\pi^2 NQTt^2}{3\delta}\right)$, applied to the problem in Equations (1a)-(1c) is $\delta$-safe and the number of constraints violations is constant in $T$.*

In the experimental section, we will see that in practical cases, the loss due to the safety requirement does not impact the performance in terms of regret too much. Conversely, in what follows, we show that when a tolerance in the violation of the constraints is accepted, an adaptation of GCB$_{\texttt{safe}}$ can be exploited to obtain a sublinear pseudo-regret.

### 5.3 GUARANTEEING SUBLINEAR PSEUDO-REGRET AND SAFETY WITH TOLERANCE: GCB$_{\texttt{safe}}(\psi, \phi)$

Given an instance of the problem in Equations (1a)-(1c) that we call *original problem*, we build an *auxiliary problem* in which we slightly relax the ROI and budget constraints. Formally, the GCB$_{\texttt{safe}}(\psi, \phi)$ is the GCB$_{\texttt{safe}}$ applied to the auxiliary problem in which the parameters $\lambda$ and $\beta$ have been substituted with $\lambda - \psi$ and $\beta + \phi$, respectively.[7] Thanks to the results in Section 5.2,

---

[7]A formal definition of the *auxiliary problem* is provided in Appendix B.4.

$GCB_{safe}(\psi, \phi)$, w.h.p., does not violate the ROI constraint of the original problem by more than the tolerance $\psi$ and the budget constraint of the original problem by more than the tolerance $\phi$.

**Theorem 8** ($GCB_{safe}(\psi, \phi)$ pseudo-regret and safety with tolerance). *Setting*

$$\psi = 2\frac{\beta_{opt} + n_{\max}}{\beta_{opt}^2}\sigma\sum_{j=1}^{N} v_j\sqrt{2\ln\left(\frac{\pi^2 NQT^3}{3\delta'}\right)} \qquad and \qquad \phi = 2N\sigma\sqrt{2\ln\left(\frac{\pi^2 NQT^3}{3\delta'}\right)},$$

*where $\delta' \leq \delta$, $GCB_{safe}(\psi, \phi)$ provides a pseudo-regret w.r.t. the optimal solution to the original problem of $\mathcal{O}\left(\sqrt{T\sum_{j=1}^{N}\gamma_{j,T}}\right)$ with probability at least $1 - \delta - \frac{\delta'}{QT^2}$, while being $\delta$-safe w.r.t. the constraints of the auxiliary problem.*

The above result states that if we allow a violation of at most $\psi$ of the ROI constraint and of $\phi$ of the budget one, the result provided in Theorem 1 can be circumvented.[8]

Notice that the magnitude of the violation $\psi$ increases linearly in the maximum number of clicks $n_{max}$ and $\sum_{j=1}^{N} v_j$, that, in turn, increases linearly in the number of sub-campaigns $N$. This suggests that in large instances, this value may be large. However, in practice, the maximum number of clicks of a sub-campaign $n_{\max}$ is a sublinear function in the optimal budget $\beta_{opt}$, and usually, it goes to a constant as the budget spent goes to infinity. Moreover, the number of sub-campaigns $N$ usually depends on the budget, *i.e.*, the budget planned by the business units is linear in the number of sub-campaigns. As a result, $\beta_{opt}$ is of the same order of $\sum_{j=1}^{N} v_j$, and therefore, since $n_{max}$ is sublinear in $\beta_{opt}$ and $\sum_{j=1}^{N} v_j$ is of the order of $\beta_{opt}$, the final expression of $\psi$ is sub-linear in $\beta_{opt}$. This means that the lower bound to $\psi$ to satisfy the assumption needed by Theorem 8 goes to zero as $\beta_{opt}$ increases.

Conversely, the most relevant dependence on the magnitude of $\phi$ is the number of campaigns $N$. This is reasonable since the more the subcampaigns, the more we have potential variance over the costs, which should be balanced with a larger violation of the constraint. This suggests that the $GCB_{safe}(\psi, \phi)$ will not be effective for large instances of the analysed optimization problem.

## 6 EXPERIMENTAL EVALUATION

We experimentally evaluate our algorithms in terms of pseudo-regret and safety in synthetic settings. The adoption of synthetic settings allows us to evaluate our algorithms in realistic scenarios and, at the same time, to have an optimal clairvoyant solution necessary to measure the algorithms' pseudo-regret and safety. In the following experiment, we show that $GCB$ suffers from significant violations of both ROI and budget constraints even in simple settings, while $GCB_{safe}$ does not.[9]

**Setting** We simulate $N = 5$ subcampaigns, with $|X_j| = 201$ bid values evenly spaced in $[0, 2]$, $|Y| = 101$ cost values evenly spaced in $[0, 100]$, and $|R| = 151$ revenue values evenly spaced in $[0, 1200]$. For a generic subcampaign $C_j$, at every $t$, the daily number of clicks is returned by the function $\tilde{n}_j(x) := \theta_j(1 - e^{-x/\delta_j}) + \xi_j^n$ and the daily cost by the function $\tilde{c}_j(x) = \alpha_j(1 - e^{-x/\gamma_j}) + \xi_j^c$, where $\theta_j \in \mathbb{R}^+$ and $\alpha_j \in \mathbb{R}^+$ represent the maximum achievable number of clicks and cost for subcampaign $C_j$ in a single day, $\delta_j \in \mathbb{R}^+$ and $\gamma_j \in \mathbb{R}^+$ characterize how fast the two functions reach a saturation point, and $\xi_j^n$ and $\xi_j^c$ are noise terms drawn from a $\mathcal{N}(0, 1)$ Gaussian distribution (these functions are customarily used in the advertising literature, *e.g.*, by Kong et al. (2018)). We assume a unitary value for each click, *i.e.*, $v_j = 1$ for each $j \in \{1, \ldots, N\}$. The values of the parameters of cost and revenue functions of the subcampaigns are specified in Table 2 reported in Appendix D.5. We set a daily budget $\beta = 100$, $\lambda = 10$ in the ROI constraint, and a time horizon $T = 60$. Notice that in this setting at the optimal solution, the budget constraint is active, while the ROI constraint is not.

For both $GCB$ and $GCB_{safe}$, the kernels for the number of clicks GPs $k(x, x')$ and for the costs GPs $h_j(x, x')$ are squared exponential kernels of the form $\sigma_f^2 \exp\left\{-\frac{(x-x')^2}{l}\right\}$ for every $x, x' \in X_j$,

---

[8]Theoretical results in settings in which we have a priori information on the looseness of either one of the constraints and we allow a violation of either one of the constraints are provided in Appendix B.4.

[9]Additional experiments and details useful for the complete reproducibility of our results are provided in Appendix D. Code available at https://github.com/oi-tech/safe_bid_opt.

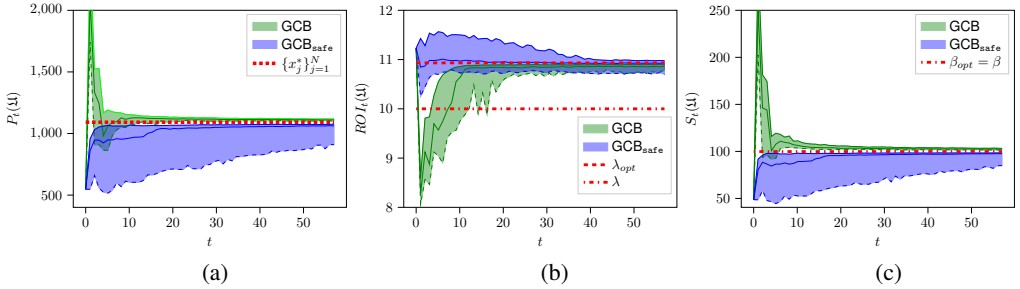

(a)  (b)  (c)

Figure 1: Daily revenue (a), ROI (b), and spend (c) obtained by GCB and GCB$_{\tt safe}$. Dashed lines correspond to the optimal values for the revenue and ROI, while dash-dotted lines correspond to the values of the ROI and budget constraints.

where the parameters $\sigma_f \in \mathbb{R}^+$ and $l \in \mathbb{R}^+$ are estimated from data, as suggested by Rasmussen & Williams (2006). The confidence for the algorithms is $\delta = 0.2$. We evaluate the algorithms in terms of daily revenue $P_t(\mathfrak{U}) := \sum_{j=1}^{N} v_j n_j(\hat{x}_{j,t})$, daily ROI: $ROI_t(\mathfrak{U}) := \frac{\sum_{j=1}^{N} v_j \, n_j(\hat{x}_{j,t})}{\sum_{j=1}^{N} c_j(\hat{x}_{j,t})}$, and daily spend: $S_t(\mathfrak{U}) := \sum_{j=1}^{N} c_j(\hat{x}_{j,t})$. We perform $100$ independent runs for each algorithm.

**Results**  In Figure 1, for the daily revenue, ROI, and spend achieved by GCB and GCB$_{\tt safe}$ at every $t$, we show the $50_{th}$ percentile (*i.e.*, the median) with solid lines and the $90_{th}$ and $10_{th}$ percentiles with dashed lines surrounding the semi-transparent area. While GCB achieves a larger revenue than GCB$_{\tt safe}$, it violates the budget constraint over the entire time horizon and the ROI constraint in the first 7 days in more than $50\%$ of the runs. This happens because, in the optimal solution, the ROI constraint is not active, while the budget constraint is. Conversely, GCB$_{\tt safe}$ satisfies the budget and ROI constraints over the time horizon for more than $90\%$ of the runs and has a slower convergence to the optimal revenue. If we focus on the median revenue, GCB$_{\tt safe}$ has a similar behaviour to that of GCB for $t > 15$. This makes GCB$_{\tt safe}$ a good choice, even in terms of overall revenue. However, it is worth noticing that, in the $10\%$ of the runs, GCB$_{\tt safe}$ does not converge to the optimal solution before the end of the learning period. These results confirm our theoretical analysis showing that limiting the exploration to safe regions might lead the algorithm to get a large regret. Furthermore, let us remark that the learning dynamics of GCB$_{\tt safe}$ are much smoother than those of GCB, which present, instead, oscillations.

## 7 CONCLUSIONS AND FUTURE WORKS

In this paper, we propose a novel framework for Internet advertising campaigns. While previous works available in the literature focus only on the maximization of the revenue provided by the campaign, we introduce the concept of *safety* for the algorithms choosing the bid allocation each day. More specifically, we want that the bidding satisfies, with high probability, some daily ROI and budget constraints fixed by the business units of the companies. Our goal is to maximize the revenue satisfying w.h.p. the uncertain constraints (a.k.a. safety). We model this setting as a combinatorial optimization problem, proving that such a problem is inapproximable within any strictly positive factor unless $\mathsf{P} = \mathsf{NP}$, but it admits an exact pseudo-polynomial-time algorithm. Most interestingly, we prove that no online learning algorithm can provide sublinear pseudo-regret while guaranteeing a sublinear number of violations of the uncertain constraints. We show that the GCB algorithm suffers from a sublinear pseudo-regret, but it may violate the constraints a linear number of times. Thus, we design GCB$_{\tt safe}$, a novel algorithm that guarantees safety at the cost of a linear pseudo-regret. Remarkably, a simple adaptation of GCB$_{\tt safe}$, namely GCB$_{\tt safe}(\psi, \phi)$, guarantees a sublinear pseudo-regret and safety at the cost of tolerances $\psi$ and $\phi$ on the ROI and budget constraints, respectively. Finally, we evaluate the empirical performance of our algorithms with synthetically advertising problems that confirmed the theoretical results provided before.

An interesting open research direction is the design of an algorithm that adopts constraints changing during the learning process, i.e., that identifies the active constraint and relaxes those that are not active. Moreover, understanding the relationship between the relaxation of one of the constraints and the increase in revenue constitutes an interesting line of research.

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

# Supplementary Material for Paper ID: 5352

## A  RELATED WORKS

Many works study Internet advertising, both from the *publisher* perspective (*e.g.*, Vazirani et al. (2007) design auctions for ads allocation and pricing) and from the *advertiser* perspective (*e.g.*, Feldman et al. (2007) study the budget optimization problem in search advertising). Only a few works deal with ROI constraints, and, to the best of our knowledge, they focus on the design of auction mechanisms. In particular, Szymanski & Lee (2006) and Borgs et al. (2007) show that ROI-based bidding heuristics can lead to cyclic behavior and reduce the allocation's efficiency, while Golrezaei et al. (2021) propose more efficient auctions with ROI constraints. The learning algorithms for daily bid optimization available in the literature address only budget constraints in the restricted case in which the platform allows the advertisers to set a daily budget limit (notice that some platforms, such as TripAdvisor and Trivago, do not even allow the setting of the daily budget limit). For instance, Zhang et al. (2012) provide an *offline* algorithm that exploits accurate models of the campaigns' performance based on low-level data rarely available to the advertisers. Feng et al. (2023) propose an online auto-bidding algorithm for a single advertiser maximizing value under the Return-on-Spend constraint. However, the constraint violation is evaluated on the cumulated value of the violation over the entire time horizon (which is a condition much weaker than ours). Nuara et al. (2018) propose an *online* learning algorithm that combines combinatorial multi-armed bandit techniques (Chen et al., 2013) with regression by Gaussian Processes (Rasmussen & Williams, 2006). This work provides no guarantees on ROI. More recent works also present pseudo-regret bounds (Nuara et al., 2022) and study subcampaigns interdependencies offline (Nuara et al., 2019). Thomaidou et al. (2014) provide a genetic algorithm for budget optimization of advertising campaigns. Ding et al. (2013) and Trovò et al. (2016) address the bid optimization problem in a single subcampaign scenario when the budget constraint is cumulative over time.

A research field strictly related to our work is learning with safe exploration and constraints subject to uncertainty. The goal is to guarantee w.h.p. the constraints satisfaction during the entire learning process. The only known results on safe exploration in multi-armed bandits address the case with continuous, convex arm spaces and convex constraints. The learner can converge to the optimal solution in these settings without violating the constraints (Moradipari et al., 2020; Amani et al., 2020). Conversely, the case with discrete and/or non-convex arm spaces or non-convex constraints, such as ours, is unexplored in the literature. We remark that some bandit algorithms address uncertain constraints where the goal is their satisfaction on average (Mannor et al., 2009; Cao & Liu, 2019). However, the per-round violation can be arbitrarily large (particularly in the early stages of the learning process), not fitting with our setting as humans could be alarmed and, thus, give up on adopting the algorithm. Moreover, several works in reinforcement learning (Hans et al., 2008; Pirotta et al., 2013; Garcia & Fernández, 2012) and multi-armed bandit (Galichet et al., 2013; Sui et al., 2015) investigate safe exploration, providing safety guarantees on the revenue provided by the algorithm, but not on the satisfaction w.h.p. of uncertain constraints.

Finally, another line of research dealing with bandits with constraints is the one related to Combinatorial MAB with knapsack constraints, e.g., the works by Sankararaman & Slivkins (2018); Badanidiyuru et al. (2018). However, they cannot be applied to our setting for two main reasons. First, they provide theoretical guarantees in the case the decision sets are matroidal-shaped. However, the set of arms in our setting is non-matroidal. More specifically, all the feasible actions $S_t = (x_{i,t}, \ldots, x_{N,t})$ have exactly $N$ elements (one per subcampaign), and removing one element from a feasible superarm generates a new superarm $S'_t = (x_{i,t}, \ldots, x_{i-1,t}, x_{i+1,t}, \ldots, x_{N,t})$ that is not an acceptable allocation of the campaign. Second, they guarantee the satisfaction of the budget constraints over a given time horizon. That is, the bandit algorithm cannot spend more than a given threshold within the given time horizon. Conversely, in our work, we bound, with high probability, the violation of the constraint in every single round.

# B  OMITTED PROOFS

## B.1  PROOFS OMITTED FROM SECTION 2

**Theorem 1** (Inapproximability). *For any $\rho \in (0,1]$, there is no polynomial-time algorithm returning a $\rho$-approximation to the problem in Equations (1a)-(1c), unless* $\mathsf{P} = \mathsf{NP}$.

*Proof.* We restrict to the instances of SUBSET-SUM such that $z \leq \sum_{j \in S} u_j$. Solving these instances is trivially NP-hard, as any instance with $z > \sum_{j \in S} u_j$ is not satisfiable, and we can decide it in polynomial time. Given an instance of SUBSET-SUM, let $\ell = \frac{\sum_{j \in S} u_j + 1}{\rho}$. Let us notice that, the lower the degree of approximation we aim, the larger the value of $\ell$. For instance, when study the problem of computing an exact solution, we set $\rho = 1$ and therefore $\ell = \sum_{j \in S} u_j + 1$, whereas, when we require a 1/2-approximation, we set $\rho = 1/2$ and therefore $\ell = 2(\sum_{j \in S} u_i + 1)$. We have $|S| + 1$ subcampaigns, each denoted with $C_j$. The available bid values belong to $\{0, 1\}$ for every subcampaign $C_j$. The parameters of the subcampaigns are set as follows.

- Subcampaign $C_0$: we set $v_0 = 1$, and
$$c_0(x) = \begin{cases} 2\ell + z & \text{if } x = 1 \\ 0 & \text{otherwise} \end{cases}, \qquad n_0(x) = \begin{cases} \ell & \text{if } x = 1 \\ 0 & \text{otherwise} \end{cases}.$$

- Subcampaign $C_j$ for every $j \in S$: we set $v_j = 1$, and
$$c_j(x) = \begin{cases} u_j & \text{if } x = 1 \\ 0 & \text{otherwise} \end{cases}, \qquad n_j(x) = \begin{cases} u_j & \text{if } x = 1 \\ 0 & \text{otherwise} \end{cases}.$$

We set the daily budget $\beta = 2(z + \ell)$ and the ROI limit $\lambda = \frac{1}{2}$.[10]

We show that if a SUBSET-SUM instance is satisfiable, then the corresponding instance of our problem admits a solution with revenue larger than $\ell$, while if a SUBSET-SUM instance is not satisfiable, the maximum revenue in the corresponding instance of our problem is at most $\rho\,\ell - 1$. Thus, the application of any polynomial-time $\rho$-approximation algorithm to instances of our problem generated from instances of SUBSET-SUM as described above would return a solution whose value is not smaller than $\rho\,\ell$ when the SUBSET-SUM instance is satisfiable, and it is not larger than $\rho\,\ell - 1$ when the SUBSET-SUM instance is not satisfiable. As a result, whenever such an algorithm returns a solution with a value that is not smaller than $\rho\,\ell$, we can decide that the corresponding SUBSET-SUM instance is satisfiable. Analogously, whenever such an algorithm returns a solution with a value that is in the range $[\rho(\rho\,\ell - 1), \rho\,\ell - 1]$, we can decide that the corresponding SUBSET-SUM instance is not satisfiable. Let us notice that the range $[\rho(\rho\,\ell - 1), \rho\,\ell - 1]$ is well defined for every $\rho \in (0, 1]$, as, by construction, $\rho\,\ell = \sum_{j \in S} u_j + 1 \geq 1$ and therefore $\rho\,\ell - 1 \geq \rho(\rho\,\ell - 1)$. Hence, such an algorithm would decide in polynomial time whether or not a SUBSET-SUM instance is satisfiable, but this is not possible unless $\mathsf{P} = \mathsf{NP}$. Since this holds for every $\rho \in (0, 1]$, then no $\rho$-approximation to our problem is allowed in polynomial time unless $\mathsf{P} = \mathsf{NP}$.

**If.** Suppose that SUBSET-SUM is satisfied by the set $S^* \subseteq S$ and that its solution assigns $x_j = 1$ if $j \in S^*$ and $x_j = 0$ otherwise, and it assigns $x_0 = 1$. The total revenue is $\ell + z \geq \ell$ and the constraints are satisfied. In particular, the sum of the costs is $2\ell + z + z = 2(\ell + z)$, while ROI $= \frac{\ell + z}{2\ell + 2z} = \frac{1}{2}$.

**Only if.** Assume by contradiction that the instance of our problem admits a solution with a revenue strictly larger than $\rho\,\ell - 1$ and that SUBSET-SUM is not satisfiable. Then, it is easy to see that we need $x_0 = 1$ for campaign $C_0$ as the maximum achievable revenue is $\sum_{j \in S} u_j = \rho\,\ell - 1$ when $x_0 = 0$. Thus, since $x_0 = 1$, the budget constraint forces $\sum_{j \in S: x_j = 1} c_i(x_j) \leq z$, thus implying $\sum_{j \in S: x_j = 1} u_j \leq z$. By the satisfaction of the ROI constraint, *i.e.*, $\frac{\sum_{j \in S: x_j = 1} u_j + l}{\sum_{j \in S: x_j = 1} u_j + 2l + z} \geq \frac{1}{2}$, it must

---

[10]For the sake of clarity, the proof uses simple instances. Adopting these instances is crucial to identify the most basic settings in which the problem is hard, and it is customarily done in the literature. Let us notice that it is possible to prove the theorem using more realistic instances. For example, we can build a reduction in which the costs are smaller than the values, *i.e.*, $c_j(x) < n_j(x)v_j$. In particular, the reduction holds even if we set $c_0(1) = \epsilon(2l + z)$, $c_j(1) = \epsilon u_j$, $\beta = 2\epsilon(z + l)$, and $\lambda = 1/(2\epsilon)$ for an arbitrary small $\epsilon$.

hold $\sum_{i \in S : x_i = 1} u_i \geq z$. Therefore, the set $S^* = \{i \in S : x_i = 1\}$ is a solution to SUBSET-SUM, thus reaching a contradiction. This concludes the proof. $\qquad\square$

### B.2 Proofs Omitted from Section 3

**Theorem 2** (Pseudo-regret/safety tradeoff)**.** *For every $\epsilon > 0$ and time horizon $T$, there is no algorithm with pseudo-regret smaller than $(1/2 - \epsilon)T$ and that violates (in expectation) the constraints less than $(1/2 - \epsilon)T$ times.*

*Proof.* In what follows, we provide an impossibility result for the optimization problem in Equations (1a)-(1c). For the sake of simplicity, our proof is based on the violation of (budget) Constraint (1c), but its extension to the violation of (ROI) Constraint (1b) is direct. Initially, we show that an algorithm satisfying the two conditions of the theorem can be used to distinguish between $\mathcal{N}(1, 1)$ and $\mathcal{N}(1 + \delta, 1)$ with an arbitrarily large probability using a number of samples independent from $\delta$.[11] Consider two instances of the bid optimization problem defined as follows. Both instances have a single subcampaign with $x \in \{0, 1\}$, $c(0) = 0$, $r(0) = 0$, $r(1) = 1$, $\beta = 1$, and $\lambda = 0$. The first instance has cost $c^1(1) = \mathcal{N}(1, 1)$, while the second one has cost $c^2(1) = \mathcal{N}(1 + \delta, 1)$. With the first instance, the algorithm must choose $x = 1$ at least $T(1/2 + \epsilon)$ times in expectation otherwise, the pseudo-regret would be strictly greater than $T(1/2 - \epsilon)$, while with the second instance, the algorithm must choose $x = 1$ at most than $T(1/2 - \epsilon)$ times in expectation. Otherwise, the constraint on the budget would be violated strictly more than $T(1/2 - \epsilon)$ times. Standard concentration inequalities imply that, for each $\gamma > 0$, there exists a $n(\epsilon, \gamma)$ such that, given $n(\epsilon, \gamma)$ runs of the learning algorithm, with the first instance the algorithm plays $x = 1$ strictly more than $Tn(\epsilon, \gamma)/2$ times with probability at least $1 - \gamma$, while with the second instance, it is played strictly less than $Tn(\epsilon, \gamma)/2$ times with probability at least $1 - \gamma$. This entails that the learning algorithm can distinguish with arbitrarily large success probability (independent of $\delta$) between the two instances using (at most) $n(\epsilon, \gamma)T$ samples from one of the normal distributions. However, the Kullback-Leibler divergence (Kullback & Leibler, 1951) between the two normal distributions is $KL(\mathcal{N}(1, 1), \mathcal{N}(1 + \delta, 1)) = \delta^2/2$ and each algorithm needs at least $\Omega(1/\delta^2)$ samples to distinguish between the two distributions with arbitrarily large probability. Since $\delta$ can be arbitrarily small, we have a contradiction. Thus, such an algorithm cannot exist. This concludes the proof.[12] $\qquad\square$

### B.3 Proofs Omitted from Section 4

**Theorem 3** (Optimality)**.** *The $\mathsf{Opt}(\boldsymbol{\mu}, \lambda)$ subroutine returns the optimal solution to the problem in Equations (1a)-(1c) when $\overline{w}_j(x) = \underline{w}_j(x) = v_j\, n_j(x)$ and $\overline{c}_j(x) = c_j(x)$ for each $j \in \{1, \dots, N\}$ and the values of revenues and costs are in $R$ and $Y$, respectively.*

*Proof.* Since all the possible values for the revenues and costs are taken into account in the subroutine, the elements in $S(y, r)$ satisfy the two inequalities in Equation (8) with the equal sign. Therefore, all the elements in $S(y, r)$ would contribute to the computation of the final value of the ROI and budget constraints, *i.e.*, the ones after evaluating all the $N$ subcampaigns, with the same values for revenue and costs, being their overall revenue equal to $r$ and their overall cost equal to $y$. Notice that Constraint (1c) is satisfied as long as it holds $\max(Y) = \beta$. The maximum operator in Line 11 excludes only solutions with the same costs and lower revenue, and, therefore, the subroutine excludes only solutions that would never be optimal (and, for this reason, said dominated). The same reasoning also holds for the subcampaign $C_1$ analysed by the algorithm. Finally, after all the dominated allocations have been discarded, the solution is selected by Equation (9), *i.e.*, among all the solutions satisfying the ROI constraints, the one with the largest revenue is selected. $\qquad\square$

---

[11]With $\mathcal{N}(a, b)$ we denote the Gaussian distribution with mean $a$ and variance $b$.

[12]Notice that the theorem can be modified to hold even with instances that satisfy real-world assumptions, *e.g.*, with costs much smaller than the budget. Indeed, we can apply the same reduction in which the costs are arbitrary, *e.g.*, $c(0) = c(1) = q$ with an arbitrary small $q$ and $\beta = 1$, while the utilities are $r(0) = 0$, $r(1) = \mathcal{N}(1, 1)$ or $r(1) = \mathcal{N}(1 - \delta, 1)$, and the ROI limit is $\lambda = 1/q$.

### B.4 OMITTED PROOFS FROM SECTION 5

**Theorem 4** (GCB pesudo-regret). *Given $\delta \in (0, 1)$, GCB applied to the problem in Equations (1a)-(1c), with probability at least $1 - \delta$, suffers from a pseudo-regret of:*

$$R_T(\text{GCB}) \leq \sqrt{\frac{8T v_{\max}^2 N^3 b_T}{\ln(1 + \sigma^2)} \sum_{j=1}^{N} \gamma_{j,T}} \ ,$$

*where $b_t := 2 \ln\left(\frac{\pi^2 NQTt^2}{3\delta}\right)$ is an uncertainty term used to guarantee the confidence level required by GCB, $v_{\max} := \max_{j \in \{1,\dots,N\}} v_j$ is the maximum value per click over all subcampaigns, and $Q := \max_{j \in \{1,\dots,N\}} |X_j|$ is the number of bids in a subcampaign.*

*Proof.* This proof extends the proof provided by Accabi et al. (2018) to the case in which multiple independent GPs are present in the optimization problem.

Let us define $r_{\boldsymbol{\mu}}(\mathbf{x})$ as the expected reward provided by a specific allocation $\mathbf{x} = (x_1, \dots, x_N)$ under the assumption that the parameter vector of the optimization problem is $\boldsymbol{\mu}$. Moreover, let

$$\boldsymbol{\eta} := \left[ w_1(x_1), \dots, w_N(x_{|X_N|}), w_1(x_1), \dots, w_N(x_{|X_N|}), -c_1(x_1), \dots, -c_N(x_{|X_N|}) \right],$$

be the vector characterizing the optimization problem in Equations (1a)-(1c), $\mathbf{x}_t$ be the allocation chosen by the GCB algorithm at round $t$, $\mathbf{x}_{\boldsymbol{\eta}}^*$ the optimal allocation—*i.e.*, the one solving the discrete version of the optimization problem in Equations (1a)-(1c) with parameter $\boldsymbol{\eta}$—, and $r_{\boldsymbol{\eta}}^*$ the corresponding expected reward.

To guarantee that GCB provides a sublinear pseudo-regret, we need a few assumptions to be satisfied. More specifically, we need a *monotonicity property*, stating that the value of the objective function increases as the values of the elements in $\boldsymbol{\mu}$ increase and a *Lipschitz continuity* assumption between the parameter vector $\boldsymbol{\mu}$ and the value returned by the objective function in Equation (1a). Formally:

**Assumption 1** (Monotonicity). *The expected reward $r_{\boldsymbol{\mu}}(S) := \sum_{j=1}^{N} v_j \, n_j(x_{j,t})$, where $S$ is the bid allocation, is monotonically non decreasing in $\boldsymbol{\mu}$, i.e., given $\boldsymbol{\mu}, \boldsymbol{\eta}$ s.t. $\mu_i \leq \eta_i$ for each $i$, we have $r_{\boldsymbol{\mu}}(S) \leq r_{\boldsymbol{\eta}}(S)$ for each $S$.*

**Assumption 2** (Lipschitz continuity). *The expected reward $r_{\boldsymbol{\mu}}(S)$ is Lipschitz continuous in the infinite norm w.r.t. the expected payoff vector $\boldsymbol{\mu}$, with Lipschitz constant $\Lambda > 0$. Formally, for each $\boldsymbol{\mu}, \boldsymbol{\eta}$ we have $|r_{\boldsymbol{\mu}}(S) - r_{\boldsymbol{\eta}}(S)| \leq \Lambda \|\boldsymbol{\mu} - \boldsymbol{\eta}\|_\infty$, where the infinite norm of a payoff vector is $\|\boldsymbol{\mu}\|_\infty := \max_i |\mu_i|$.*

Our problem satisfies both of the above assumptions. Indeed, we have that the Lipschitz continuity holds with constant $\Lambda = v_{\max} N$. Instead, the monotonicity property holds by definition of $\boldsymbol{\mu}$, as the increase of a value of $\overline{w}_j(x)$ would increase the value of the objective function, and the increase of the values of $\underline{w}_j(x)$ or $\overline{c}_j(x)$ would enlarge the feasibility region of the problem, thus not excluding optimal solutions.

Let us now focus on the per-step expected regret, defined as:

$$reg_t := r_{\boldsymbol{\eta}}^* - r_{\boldsymbol{\eta}}(\mathbf{x}_t).$$

Let us recall a property of the Gaussian distribution which will be useful in what follows. Be $r \sim \mathcal{N}(0, 1)$ and $c \in \mathbb{R}^+$, we have:

$$\mathbb{P}[r > c] = \frac{1}{\sqrt{2\pi}} e^{-\frac{c^2}{2}} \int_c^\infty e^{-\frac{(r-c)^2}{2} - c(r-c)} \, dr$$

$$\leq e^{-\frac{c^2}{2}} \mathbb{P}[r > 0] = \frac{1}{2} e^{-\frac{c^2}{2}},$$

since $e^{-c(r-c)} \leq 1$ for $r \geq c$. For the symmetry of the Gaussian distribution, we have:

$$\mathbb{P}[|r| > c] \leq e^{-\frac{c^2}{2}}. \tag{10}$$

Let us focus on the GP modeling the number of clicks. Following Lemma 5 in the work by Srinivas et al. (2010), we have that conditioned on the number of clicks $(\tilde{n}_{j,1}(x_{j,1}), \dots, \tilde{n}_{j,t}(x_{j,t}))$, the selected bids $(x_{j,1}, \dots, x_{j,1})$, with $x_{j,h} \in X_j$, are deterministic and the estimated number of clicks follows:

$$n_{j,t}(x) \sim \mathcal{N}(\hat{n}_{j,t}(x), (\hat{\sigma}_{j,t}^n(x))^2),$$

for all $x \in X_j$. Thus, substituting $r = \frac{\hat{n}_{j,t}(x) - n_{j,t}(x)}{\hat{\sigma}^n_{j,t}(x)}$ and $c = \sqrt{b_t}$ in Equation (10), we obtain:

$$\mathbb{P}\left[|\hat{n}_{j,t}(x) - n_{j,t}(x)| > \sqrt{b_t}\hat{\sigma}^n_{j,t}(x)\right] \leq e^{-\frac{b_t}{2}}. \tag{11}$$

Recall that, after $n$ rounds, each arm can be chosen a number of times from 1 to $n$. Applying the union bound over the rounds ($h \in \{1, \ldots, T\}$), the sub-campaigns $C_j$ ($C_j$ with $j \in \{1, \ldots, N\}$), the number of times the arms in $C_j$ are chosen ($t \in \{1, \ldots, n\}$), and the available arms in $C_j$ ($x \in X_j$), and exploiting Equation (11), we obtain:

$$\mathbb{P}\left[\bigcup_{h,j,t,x}\left(|\hat{n}_{j,t}(x) - n_{j,t}(x)| > \sqrt{b_t}\hat{\sigma}^n_{j,t}(x)\right)\right] \tag{12}$$

$$\leq \sum_{h=1}^{T}\sum_{j=1}^{N}\sum_{t=1}^{n} |X_j| e^{-\frac{b_t}{2}}. \tag{13}$$

Thus, choosing $b_t = 2\ln\left(\frac{\pi^2 NQTt^2}{3\delta}\right)$, we obtain:

$$\sum_{h=1}^{T}\sum_{j=1}^{N}\sum_{t=1}^{n} |X_j| e^{-\frac{b_t}{2}} \leq \sum_{h=1}^{T}\sum_{j=1}^{N}\sum_{t=1}^{n} Q\frac{3\delta}{\pi^2 NQTt^2}$$

$$\sum_{n=1}^{\infty}\frac{2\delta}{\pi^2 t^2} = \frac{\delta}{2},$$

where we used the fact that $Q \geq |X_j|$ for each $j \in \{1, \ldots N\}$.

Using the same proof on the GP defined over the costs leads to:

$$\mathbb{P}\left[\bigcup_{h,j,t,x}\left(|\hat{c}_{j,t}(x) - c_{j,t}(x)| > \sqrt{b_t}\hat{\sigma}^c_{j,t}(x)\right)\right] \leq \frac{\delta}{2}.$$

The above proof implies that the union of the event that all the bounds used in the GCB algorithm holds with probability at least $1 - \delta$. Formally, for each $t \geq 1$, we know that with probability at least $1 - \delta$ the following holds for all $x_j \in X_j$, $j \in \{1, \ldots N\}$, and number of times the the arm $x_j$ has been pulled over $t$ rounds:

$$|\hat{n}_j(x_j) - n_j(x_j)| \leq \sqrt{b_t}\hat{\sigma}^n_{j,t}(x_j), \tag{14}$$

$$|\hat{c}_j(x_j) - c_j(x_j)| \leq \sqrt{b_t}\hat{\sigma}^c_{j,t}(x_j). \tag{15}$$

From now on, let us assume we are in the *clean event* that the previous bounds hold.

Let us focus on the term $r_{\boldsymbol{\mu}}(\mathbf{x}_t)$. The following holds:

$$r_{\boldsymbol{\mu}}(\mathbf{x}_t) \geq r^*_{\boldsymbol{\mu}} \geq r_{\boldsymbol{\mu}}(\mathbf{x}^*_{\boldsymbol{\mu}}) \geq r_{\boldsymbol{\eta}}(\mathbf{x}^*_{\boldsymbol{\mu}}) = r^*_{\boldsymbol{\eta}}, \tag{16}$$

where we use the definition of $r^*_{\boldsymbol{\mu}}$, and the monotonicity property of the expected reward (Assumption 1), being $(\boldsymbol{\mu})_i \geq (\boldsymbol{\eta})_i, \forall i$. Using Equation (16), the instantaneous expected pseudo-regret $reg_t$ at round $t$ satisfies the following inequality:

$$reg_t = r^*_{\boldsymbol{\eta}} - r_{\boldsymbol{\eta}}(\mathbf{x}_t) \leq r_{\boldsymbol{\mu}}(\mathbf{x}_t) - r_{\boldsymbol{\eta}}(\mathbf{x}_t) = \tag{17}$$

$$\leq \underbrace{r_{\boldsymbol{\mu}}(\mathbf{x}_t) - r_{\hat{\boldsymbol{\mu}}}(\mathbf{x}_t)}_{r_a} + \underbrace{r_{\hat{\boldsymbol{\mu}}}(\mathbf{x}_t) - r_{\boldsymbol{\eta}}(\mathbf{x}_t)}_{r_b}, \tag{18}$$

where

$$\hat{\boldsymbol{\mu}} := \left[\hat{w}_{1,t-1}(x_1), \ldots, \hat{w}_{N,t-1}(x_{|X_N|}), \hat{w}_{1,t-1}(x_1), \ldots, \hat{w}_{N,t-1}(x_{|X_N|}),\right. \tag{19}$$

$$\left. -\hat{c}_{1,t-1}(x_1), \ldots, -\hat{c}_{N,t-1}(x_{|X_N|})\right],$$

is the vector composed of the estimated average payoffs for each arm $x \in X_j$ and each campaign $C_j$, where $\hat{w}_{j,t-1}(x) := v_j\hat{n}_{j,t-1}(x)$.

We use the Lipschitz property of the expected reward function (see Assumption 2) to bound the terms in Equation (18) as follows:

$$r_a \leq \Lambda\|\boldsymbol{\mu} - \hat{\boldsymbol{\mu}}\|_\infty = \Lambda \max_{j \in \{1, \ldots, N\}}\left(v_{\max}\sqrt{b_t} \max_{x \in X_j} \hat{\sigma}^n_{j,t}(x)\right) \tag{20}$$

$$\leq Nv_{\max}\sqrt{b_t} \max_{j \in \{1, \ldots, N\}}\left(\max_{x \in X_j} \hat{\sigma}^n_{j,t}(x)\right) \tag{21}$$

$$\leq N v_{\max} \sqrt{b_t} \sum_{j=1}^{N} \left( \max_{x \in X_j} \hat{\sigma}_{j,t}^n(x) \right), \tag{22}$$

$$r_b \leq \Lambda \|\hat{\boldsymbol{\mu}} - \boldsymbol{\eta}\|_\infty$$

$$\leq N v_{\max} \sqrt{b_t} \sum_{j=1}^{N} \left( \max_{x \in X_j} \hat{\sigma}_{j,t}^n(x) \right), \tag{23}$$

where Equation (20) holds by the definition of $\boldsymbol{\mu}$, Equation (22) holds since the maximum over a set is not greater than the sum of the elements of the set, if they are all non-negative, and Equation (23) directly follows from Equation (14). Plugging Equations (22) and (23) into Equation (18), we obtain:

$$reg_t \leq 2 N v_{\max} \sqrt{b_t} \sum_{j=1}^{N} \left( \max_{x \in X_j} \hat{\sigma}_{j,t}^n(x) \right). \tag{24}$$

We need now to upper bound $\hat{\sigma}_{j,t}^n(x)$. Consider a realization $n_j(\cdot)$ of a GP over $X_j$ and recall that, thanks to Lemma 5.3 in (Srinivas et al., 2010), under the Gaussian assumption we can express the information gain $IG_{j,t}$ provided by $(\tilde{n}_j(\hat{x}_{j,1}), \ldots, \tilde{n}_j(\hat{x}_{j,|X_j|}))$ corresponding to the sequence of arms $(\hat{x}_{j,1}, \ldots, \hat{x}_{j,|X_j|})$ as:

$$IG_{j,t} = \frac{1}{2} \sum_{h=1}^{t} \log \left( 1 + \sigma^{-2} \left( \hat{\sigma}_{j,t}^n(\hat{x}_{j,h}) \right)^2 \right). \tag{25}$$

We have that:

$$\left( \hat{\sigma}_{j,t}^n(\hat{x}_{j,h}) \right)^2 = \sigma^2 \left[ \sigma^{-2} (\hat{\sigma}_{j,t}^n(\hat{x}_{j,h}))^2 \right] \leq \frac{\log \left[ 1 + \sigma^{-2} (\hat{\sigma}_{j,t}^n(\hat{x}_{j,h}))^2 \right]}{\log (1 + \sigma^{-2})}, \tag{26}$$

since $s^2 \leq \frac{\sigma^{-2} \log (1+s^2)}{\log(1+\sigma^{-2})}$ for all $s \in [0, \sigma^{-1}]$, and $\sigma^{-2}(\hat{\sigma}_{j,t}^n(\hat{x}_{j,h}))^2 \leq \sigma^{-2} k(\hat{x}_{j,h}, \hat{x}_{j,h}) \leq \sigma^{-2}$, where $k(\cdot, \cdot)$ is the kernel of the GP. Since Equation (26) holds for any $x \in X_j$ and for any $j \in \{1, \ldots N\}$, then it also holds for the arm $\hat{x}_{\max}$ maximizing the variance $(\hat{\sigma}_{j,t}^n(\hat{x}_{j,h}))^2$ over $X_j$. Thus, setting $\bar{c} = \frac{8 N^2}{\log(1+\sigma^{-2})}$ and exploiting the Cauchy-Schwarz inequality, we obtain:

$$\mathcal{R}_T^2(GCB) \leq T \sum_{t=1}^{T} reg_t^2$$

$$\leq T \sum_{t=1}^{T} 4 N^2 v_{\max}^2 b_t \left[ \sum_{j=1}^{N} \left( \max_{x \in X_j} \hat{\sigma}_{j,t}^n(x) \right) \right]^2$$

$$\leq 4 N^2 v_{\max}^2 T b_T \sum_{t=1}^{T} \left[ N \sum_{j=1}^{N} \max_{x \in X_j} (\hat{\sigma}_{j,t}^n(x))^2 \right]$$

$$\leq \bar{c} N v_{\max}^2 T b_T \sum_{j=1}^{N} \frac{1}{2} \sum_{t=1}^{T} \max_{x \in X_j} \log \left( 1 + \sigma^{-2} \left( \hat{\sigma}_{j,t}^n(\hat{x}_{j,h}) \right)^2 \right)$$

$$\leq \bar{c} N v_{\max}^2 T b_T \sum_{j=1}^{N} \gamma_{j,T}.$$

We conclude the proof by taking the square root on both the r.h.s. and the l.h.s. of the last inequality. $\qquad \square$

**Theorem 5** (GCB safety). *Given $\delta \in (0, 1)$, GCB applied to the problem in Equations (1a)-(1c) with $b_t := 2 \ln \left( \frac{\pi^2 NQTt^2}{3\delta} \right)$ is $\eta$-safe where $\eta \geq T - \frac{\delta}{2NQT}$ and, therefore, the number of constraints violations is linear in $T$.*

*Proof.* Let us focus on a specific day $t$. Consider the case in which Constraints (1b) and (1c) are active, and, therefore, the left side equals the right side: $\sum_{j=1}^{N} \underline{w}_j(x_{j,t}) - \lambda \sum_{j=1}^{N} \bar{c}_j(x_{j,t}) = 0$ and $\sum_{j=1}^{N} \bar{c}_j(x_{j,t}) = \beta$. For the sake of simplicity, we focus on the costs $\bar{c}_j(x_{j,t})$, but similar arguments also apply to the revenues $\underline{w}_j(x_{j,t})$. A necessary condition for which the two constraints are valid

also for the actual (non-estimated) revenues and costs is that for at least one of the costs it holds $c_j(x_{j,t}) \leq \bar{c}_j(x_{j,t})$. Indeed, if the opposite holds, *i.e.*, $\bar{c}_j(x_{j,t}) < c_j(x_{j,t})$ for each $j \in \{1, \ldots, N\}$ and $x_{j,t} \in X_j$, the budget constraint would be violated by the allocation since $\sum_{j=1}^{N} c_j(x_{j,t}) > \sum_{j=1}^{N} \bar{c}_j(x_{j,t}) = \beta$. Since the event $c_j(x_{j,t}) \leq \bar{c}_j(x_{j,t})$ occurs with probability at most $\frac{3\delta}{\pi^2 NQTt^2}$, over the $t \in \mathbb{N}$, formally:

$$\mathbb{P}\left( \frac{\sum_{j=1}^{N} v_j \, n_j(\hat{x}_{j,t})}{\sum_{j=1}^{N} c_j(\hat{x}_{j,t})} < \lambda \vee \sum_{j=1}^{N} c_j(\hat{x}_{j,t}) > \beta \right) \geq 1 - \frac{3\delta}{\pi^2 NQTt^2}.$$

Finally, summing over the time horizon $T$ the probability that the constraints are not violated is at most $\frac{\delta}{2NQT}$, formally:

$$\sum_{t=1}^{T} \mathbb{P}\left( \frac{\sum_{j=1}^{N} v_j \, n_j(\hat{x}_{j,t})}{\sum_{j=1}^{N} c_j(\hat{x}_{j,t})} < \lambda \vee \sum_{j=1}^{N} c_j(\hat{x}_{j,t}) > \beta \right) \geq T - \frac{\delta}{2NQT}.$$

This concludes the proof. $\qquad\square$

**Theorem 9** (GCB cumulated violation). *The cumulated violation of the two constraints provided by the GCB algorithm satisfies:*

- $\sum_{t=1}^{T} \sum_{j=1}^{N} c_j(x_{j,t}) - T\beta \leq \mathcal{O}\left( \sqrt{T \sum_{j=1}^{N} \gamma_{j,T}^c} \right),$

- $T\lambda - \sum_{t=1}^{T} \frac{\sum_{j=1}^{N} v_j n_j(x_{j,t})}{\sum_{j=1}^{N} c_j(x_{j,t})} \leq \mathcal{O}\left( \sqrt{T \sum_{j=1}^{N}(\gamma_{j,t} + \gamma_{j,t}^c)} \right),$

*where $\gamma_{j,t}^c$ is the maximum information gain of the GPs modeling the costs of $j$-th subcampaign after $t$ samples.*

*Proof.* We analyse the violation of the ROI constraint $vr_t$ at a specific day $t$ and the one of the budget constraint $vb_t$.

Focusing on the budget constraint, we have:

$$vb_t = \sum_{j=1}^{N} c_j(x_{j,t}) - y \leq \sum_{j=1}^{N} (\hat{c}_j(x_{j,t}) + \sqrt{b_{t-1}} \hat{\sigma}_{j,t-1}^c(x_{j,t})) - \beta \qquad (27)$$

$$= \underbrace{\sum_{j=1}^{N} (\hat{c}_j(x_{j,t}) - \sqrt{b_{t-1}} \hat{\sigma}_{j,t-1}^c(x_{j,t})) - \beta}_{\leq 0} + 2\sum_{j=1}^{N} \sqrt{b_{t-1}} \hat{\sigma}_{j,t-1}^c(x_{j,t}) \qquad (28)$$

$$\leq 2\sum_{j=1}^{N} \sqrt{b_{t-1}} \hat{\sigma}_{j,t-1}^c(x_{j,t}), \qquad (29)$$

where the inequality in Equation (28) holds from the fact that the solution selected by GCB has to satisfy the budget constraint. Define $\bar{n}_j(x_{j,t}) := \hat{n}_j(x_{j,t}) + \sqrt{b_{t-1}} \hat{\sigma}_j^n(x_{j,t})$. Notice that the previous bound holds w.p. at least $1 - \delta$ since this is the probability for which the bounds on the number of clicks and the costs hold.

Since we have $\lambda \leq \frac{\sum_{j=1}^{N} v_j \bar{n}_j(x_{j,t})}{\sum_{j=1}^{N} \bar{c}_j(x_{j,t})}$:

$$vr_t = \lambda - \frac{\sum_{j=1}^{N} v_j n_j(x_{j,t})}{\sum_{j=1}^{N} c_j(x_{j,t})} \leq \frac{\sum_{j=1}^{N} v_j \bar{n}_j(x_{j,t})}{\sum_{j=1}^{N} \bar{c}_j(x_{j,t})} - \frac{\sum_{j=1}^{N} v_j n_j(x_{j,t})}{\sum_{j=1}^{N} c_j(x_{j,t})} \qquad (30)$$

$$\leq \frac{\sum_{j=1}^{N} c_j(x_{j,t}) \sum_{j=1}^{N} v_j \bar{n}_j(x_{j,t}) - \sum_{j=1}^{N} \bar{c}_j(x_{j,t}) \sum_{j=1}^{N} v_j n_j(x_{j,t})}{\sum_{j=1}^{N} c_j(x_{j,t}) \sum_{j=1}^{N} \bar{c}_j(x_{j,t})} \qquad (31)$$

$$\leq \frac{1}{N^2 c_{\min}(c_{\min} - \sqrt{b_T}\sigma)} \left( \sum_{j=1}^{N} c_j(x_{j,t}) \sum_{j=1}^{N} v_j \bar{n}_j(x_{j,t}) - \sum_{j=1}^{N} c_j(x_{j,t}) \sum_{j=1}^{N} v_j n_j(x_{j,t}) \right)$$

$$+ \sum_{j=1}^{N} c_j(x_{j,t}) \sum_{j=1}^{N} v_j n_j(x_{j,t}) - \sum_{j=1}^{N} \overline{c}_j(x_{j,t}) \sum_{j=1}^{N} v_j n_j(x_{j,t}) \Bigg) \tag{32}$$

$$\leq \frac{1}{N^2 c_{\min}(c_{\min} - \sqrt{b_T}\sigma)} \left[ \sum_{j=1}^{N} c_j(x_{j,t}) \left( \sum_{j=1}^{N} v_j \overline{n}_j(x_{j,t}) - \sum_{j=1}^{N} v_j n_j(x_{j,t}) \right) \right.$$

$$\left. + \sum_{j=1}^{N} v_j n_j(x_{j,t}) \left( \sum_{j=1}^{N} c_j(x_{j,t}) - \sum_{j=1}^{N} \overline{c}_j(x_{j,t}) \right) \right] \tag{33}$$

$$\leq \frac{N c_{\max} v_{\max} 2 \sum_{j=1}^{N} \sqrt{b_{t-1}} \hat{\sigma}_j^n(x_{j,t}) + N n_{\max} v_{\max} 2 \sum_{j=1}^{N} \sqrt{b_{t-1}} \hat{\sigma}_j^c(x_{j,t})}{N^2 c_{\min}(c_{\min} - \sqrt{b_T}\sigma)} \tag{34}$$

$$= \frac{2 c_{\max} v_{\max} \sum_{j=1}^{N} \sqrt{b_{t-1}} \hat{\sigma}_j^n(x_{j,t}) + 2 n_{\max} v_{\max} \sum_{j=1}^{N} \sqrt{b_{t-1}} \hat{\sigma}_j^c(x_{j,t})}{N c_{\min}(c_{\min} - \sqrt{b_T}\sigma)}, \tag{35}$$

where $\sum_{j=1}^{N} v_j \hat{n}_j(x_{j,t}) \geq \sum_{j=1}^{N} v_j n_j(x_j^*)$ by definition of the GCB selection rule, $v_{\max} := \max_{j=1}^{N} v_j$, and we assume that $c_{\min} - \sqrt{b_T}\sigma > 0$.

Using arguments similar to what has been used to bound the instantaneous regret $r_t$ in Srinivas et al. (2010) and Accabi et al. (2018), and summing over the time horizon $T$, provides the final statement of the theorem. $\qquad \square$

**Theorem 6** (GCB$_{\tt safe}$ pseudo-regret). *Given $\delta \in (0, 1)$, GCB$_{\tt safe}$ with $b_t := 2\ln\left(\frac{\pi^2 NQTt^2}{3\delta}\right)$, applied to the problem in Equations (1a)-(1c) suffers from a pseudo-regret $R_t(\text{GCB}_{\tt safe}) = \Theta(T)$.*

*Proof.* At the optimal solution, at least one of the constraints is active, *i.e.,* it has the left-hand side equal to the right-hand side. Assume that the optimal clairvoyant solution $\{x_j^*\}_{j=1}^{N}$ to the optimization problem has a value of the ROI $\lambda_{opt}$ equal to $\lambda$. We showed in the proof of Theorem 7 that for any allocation, with probability at least $1 - \frac{3\delta}{\pi^2 NQTt^2}$, it holds that $\frac{\sum_{j=1}^{N} v_j\, n_j(x_{j,t})}{\sum_{j=1}^{N} c_j(x_{j,t})} > \frac{\sum_{j=1}^{N} v_j\, \underline{n}_j(x_{j,t})}{\sum_{j=1}^{N} \overline{c}_j(x_{j,t})}$. This is true also for the optimal clairvoyant solution $\{x_j^*\}_{j=1}^{N}$, for which $\lambda = \frac{\sum_{j=1}^{N} v_j\, n_j(x^*)}{\sum_{j=1}^{N} c_j(x^*)} > \frac{\sum_{j=1}^{N} v_j\, \underline{n}_j(x^*)}{\sum_{j=1}^{N} \overline{c}_j(x^*)}$, implying that the values used in the ROI constraint make this allocation not feasible for the $\text{Opt}(\boldsymbol{\mu}, \lambda)$ procedure. As shown before, this happens with probability at least $1 - \frac{3\delta}{\pi^2 NQTt^2}$ at day $t$, and $1 - \delta$ over the time horizon $T$. To conclude, with probability $1 - \delta$, not depending on the time horizon $T$, we will not choose the optimal arm during the time horizon and, therefore, the regret of the algorithm cannot be sublinear. Notice that the same line of proof is also holding in the case the budget constraint is active, therefore, the previous result holds for each instance of the problem in Equations (1a)-(1c). $\qquad \square$

**Theorem 7** (GCB$_{\tt safe}$ safety). *Given $\delta \in (0, 1)$, GCB$_{\tt safe}$ with $b_t := 2\ln\left(\frac{\pi^2 NQTt^2}{3\delta}\right)$, applied to the problem in Equations (1a)-(1c) is $\delta$-safe and the number of constraints violations is constant in $T$.*

*Proof.* Let us focus on a specific day $t$. Constraints (1b) and (1c) are satisfied by the solution of $\text{Opt}(\boldsymbol{\mu}, \lambda)$ for the properties of the optimization procedure. Define $\underline{n}_j(x_{j,t}) := \hat{n}_j(x_{j,t}) - \sqrt{b_{t-1}}\hat{\sigma}_j^n(x_{j,t})$. Thanks to the specific construction of the upper bounds, we have that $c_j(x_{j,t}) \leq \overline{c}_j(x_{j,t})$ and $n_j(x_{j,t}) \geq \underline{n}_j(x_{j,t})$, each holding with probability at least $1 - \frac{3\delta}{\pi^2 NQTt^2}$. Therefore, we have:

$$\frac{\sum_{j=1}^{N} v_j\, n_j(x_{j,t})}{\sum_{j=1}^{N} c_j(x_{j,t})} > \frac{\sum_{j=1}^{N} v_j\, \underline{n}_j(x_{j,t})}{\sum_{j=1}^{N} \overline{c}_j(x_{j,t})} \geq \lambda$$

and

$$\sum_{j=1}^{N} c_j(x_{j,t}) < \sum_{j=1}^{N} \overline{c}_j(x_{j,t}) \leq \beta.$$

Using a union bound over:

- the two GPs (number of clicks and costs);

- the time horizon $T$;

- the number of times each bid is chosen in a subcampaign (at most $t$);

- the number of arms present in each subcampaign ($|X_j|$);

- the number of subcampaigns ($N$);

we have:

$$\sum_{t=1}^{T} \mathbb{P}\left(\frac{\sum_{j=1}^{N} v_j \, n_j(\hat{x}_{j,t})}{\sum_{j=1}^{N} c_j(\hat{x}_{j,t})} < \lambda \vee \sum_{j=1}^{N} c_j(\hat{x}_{j,t}) > \beta\right) \leq 2 \sum_{j=1}^{N} \sum_{k=1}^{|X_j|} \sum_{h=1}^{T} \sum_{l=1}^{t} \frac{3\delta}{\pi^2 NQTl^2} \tag{36}$$

$$\leq 2 \sum_{j=1}^{N} \sum_{k=1}^{Q} \sum_{h=1}^{T} \sum_{l=1}^{+\infty} \frac{3\delta}{\pi^2 NQTl^2} = \delta. \tag{37}$$

This concludes the proof. $\qquad\square$

The definition of the auxiliary problem that $\mathsf{GCB}_{\mathtt{safe}}(\psi, \phi)$ algorithm is solving is the following:

$$\max_{(x_{1,t},\ldots,x_{N,t})\in X_1\times\ldots\times X_N} \sum_{j=1}^{N} v_j \, n_j(x_{j,t}) \tag{38a}$$

$$\text{s.t.} \qquad \frac{\sum_{j=1}^{N} v_j \, n_j(x_{j,t})}{\sum_{j=1}^{N} c_j(x_{j,t})} \geq \lambda - \psi, \tag{38b}$$

$$\sum_{j=1}^{N} c_j(x_{j,t}) \leq \beta + \phi. \tag{38c}$$

**Theorem 10** ($\mathsf{GCB}_{\mathtt{safe}}(\psi, 0)$ pseudo-regret and safety with tolerance)**.** *When:*

$$\psi \geq 2\frac{\beta_{opt} + n_{\max}}{\beta_{opt}^2} \sum_{j=1}^{N} v_j \sqrt{2\ln\left(\frac{\pi^2 NQT^3}{3\delta'}\right)}\sigma \qquad \text{and} \qquad \beta_{opt} < \beta \frac{\sum_{j=1}^{N} v_j}{\frac{N\,\beta_{opt}\psi}{\beta_{opt}+n_{\max}} + \sum_{j=1}^{N} v_j},$$

*where $\delta' \leq \delta$, $\beta_{opt}$ is the spend at the optimal solution of the original problem, and $n_{\max} := \max_{j,x} n_j(x)$ is the maximum over the sub-campaigns and the admissible bids of the expected number of clicks, $\mathsf{GCB}_{\mathtt{safe}}(\psi, 0)$ provides a pseudo-regret w.r.t. the optimal solution to the original problem of $\mathcal{O}\left(\sqrt{T \sum_{j=1}^{N} \gamma_{j,T}}\right)$ with probability at least $1 - \delta - \frac{\delta'}{QT^2}$, while being $\delta$-safe w.r.t. the constraints of the auxiliary problem.*

*Proof.* In what follows, we show that, at a specific day $t$, since the optimal solution of the original problem $\{x_j^*\}_{j=1}^{N}$ is included in the set of feasible ones, we are in a setting analogous to the one of $\mathsf{GCB}$, in which the regret is sublinear. Let us assume that the upper bounds on all the quantities (number of clicks and costs) holds. This has been shown before to occur with overall probability $\delta$ over the whole time horizon $T$. Moreover, notice that combining the properties of the budget of the optimal solution of the original problem $\beta_{opt}$ and using $\psi = 2\frac{\beta_{opt}+n_{\max}}{\beta_{opt}^2}\sum_{j=1}^{N} v_j \sqrt{2\ln\left(\frac{\pi^2 NQT^3}{3\delta'}\right)}\sigma$, we have:

$$\beta_{opt} < \beta \frac{\sum_{j=1}^{N} v_j}{\frac{N\,\beta_{opt}\psi}{\beta_{opt}+n_{\max}} + \sum_{j=1}^{N} v_j} \tag{39}$$

$$\left(\frac{N\,\beta_{opt}\psi}{\beta_{opt} + n_{\max}} + \sum_{j=1}^{N} v_j\right)\beta_{opt} < \beta \sum_{j=1}^{N} v_j \tag{40}$$

$$2N \sum_{j=1}^{N} v_j \sqrt{2 \ln \left( \frac{\pi^2 NQT^3}{3\delta'} \right)} \sigma + \sum_{j=1}^{N} v_j \beta_{opt} < \beta \sum_{j=1}^{N} v_j \qquad (41)$$

$$\beta > \beta_{opt} + 2N \sqrt{2 \ln \left( \frac{\pi^2 NQT^3}{3\delta'} \right)} \sigma. \qquad (42)$$

First, let us evaluate the probability that the optimal solution is not feasible. This occurs if its bounds are either violating the ROI or budget constraints. First, we show that analysing the budget constraint, the optimal solution of the original problem is feasible with high probability. Formally, it is not feasible with probability:

$$\mathbb{P} \left( \sum_{j=1}^{N} \bar{c}_j(x_j^*) > \beta \right) \leq \mathbb{P} \left( \sum_{j=1}^{N} \bar{c}_j(x_j^*) > \beta_{opt} + 2N \sqrt{2 \ln \left( \frac{\pi^2 NQT^3}{3\delta'} \right)} \sigma \right) \qquad (43)$$

$$= \mathbb{P} \left( \sum_{j=1}^{N} \bar{c}_j(x_j^*) > \sum_{j=1}^{N} c_j(x_j^*) + 2N \sqrt{2 \ln \frac{\pi^2 NQT^3}{3\delta'}} \sigma \right) \qquad (44)$$

$$\leq \sum_{j=1}^{N} \mathbb{P} \left( \bar{c}_j(x_j^*) > c_j(x_j^*) + 2 \sqrt{2 \ln \frac{\pi^2 NQT^3}{3\delta'}} \sigma \right) \qquad (45)$$

$$= \sum_{j=1}^{N} \mathbb{P} \left( \hat{c}_{j,t-1}(x_j^*) - c_j(x_j^*) > -\sqrt{b_t} \hat{\sigma}_{j,t-1}^c(x_j^*) + 2 \sqrt{2 \ln \frac{\pi^2 NQT^3}{3\delta'}} \sigma \right) \qquad (46)$$

$$\leq \sum_{j=1}^{N} \mathbb{P} \left( \hat{c}_{j,t-1}(x_j^*) - c_j(x_j^*) > \sqrt{2 \ln \frac{\pi^2 NQT^3}{3\delta'}} \hat{\sigma}_{j,t-1}^c(x_j^*) \right) \qquad (47)$$

$$\leq \sum_{j=1}^{N} \mathbb{P} \left( \frac{\hat{c}_{j,t-1}(x_j^*) - c_j(x_j^*)}{\hat{\sigma}_{j,t-1}^c(x_j^*)} > \sqrt{2 \ln \frac{\pi^2 NQT^3}{3\delta'}} \right) \qquad (48)$$

$$\leq \sum_{j=1}^{N} \frac{3\delta'}{\pi^2 NQT^3} = \frac{3\delta'}{\pi^2 QT^3}, \qquad (49)$$

where, in the inequality in Equation (43) we used Equation (42), in Equation (48) we used the fact that $\frac{\pi^2 NQt^2 T}{3\delta} \leq \frac{\pi^2 NQT^3}{3\delta'}$ for each $t \in \{1, \dots, T\}$, $\hat{\sigma}_{j,t-1}^c(x_j^*) \leq \sigma$ for each $j$ and $t$, and the inequality in Equation (49) is from Srinivas et al. (2010). Summing over the time horizon $T$, we get that the optimal solution of the original problem $\{x_j^*\}_{j=1}^{N}$ is excluded from the set of the feasible ones with probability at most $\frac{3\delta'}{\pi^2 QT^2}$.

Second, we derive a bound over the probability that the optimal solution of the original problem is feasible due to the newly defined ROI constraint. Let us notice that since the ROI constraint is active we have $\lambda = \lambda_{opt}$. The probability that $\{x_j^*\}_{j=1}^{N}$ is not feasible due to the ROI constraint is:

$$\mathbb{P} \left( \frac{\sum_{j=1}^{N} v_j \, \underline{n}_j(x_j^*)}{\sum_{j=1}^{N} \bar{c}_j(x_j^*)} < \lambda - \psi \right) \qquad (50)$$

$$\leq \mathbb{P} \left( \frac{\sum_{j=1}^{N} v_j \, \underline{n}_j(x_j^*)}{\sum_{j=1}^{N} \bar{c}_j(x_j^*)} < \lambda_{opt} - 2 \frac{\beta_{opt} + n_{\max}}{\beta_{opt}^2} \sum_{j=1}^{N} v_j \sqrt{2 \ln \frac{\pi^2 NQT^3}{3\delta'}} \sigma \right) \qquad (51)$$

$$= \mathbb{P} \left( \frac{\sum_{j=1}^{N} v_j \, \underline{n}_j(x_j^*)}{\sum_{j=1}^{N} \bar{c}_j(x_j^*)} < \frac{\sum_{j=1}^{N} v_j \, n_j(x_j^*)}{\sum_{j=1}^{N} c_j(x_j^*)} - 2 \frac{\beta_{opt} + n_{\max}}{\beta_{opt}^2} \sum_{j=1}^{N} v_j \sqrt{2 \ln \frac{\pi^2 NQT^3}{3\delta'}} \sigma \right) \qquad (52)$$

$$= \mathbb{P} \left( \sum_{j=1}^{N} c_j(x_j^*) \sum_{j=1}^{N} v_j \, \underline{n}_j(x_j^*) < \sum_{j=1}^{N} \bar{c}_j(x_j^*) \sum_{j=1}^{N} v_j \, n_j(x_j^*) \right)$$

$$-2\frac{\beta_{opt}+n_{\max}}{\beta_{opt}^2}\sum_{j=1}^N c_j(x_j^*)\sum_{j=1}^N \bar{c}_j(x_j^*)\sum_{j=1}^N v_j\sqrt{2\ln\frac{\pi^2 NQT^3}{3\delta'}}\sigma\Bigg) \tag{53}$$

$$=\mathbb{P}\Bigg(\sum_{j=1}^N c_j(x_j^*)\sum_{j=1}^N v_j\,\underline{n}_j(x_j^*)-\sum_{j=1}^N c_j(x_j^*)\sum_{j=1}^N v_j\,n_j(x_j^*)+$$

$$\frac{2}{\beta_{opt}}\sum_{j=1}^N c_j(x_j^*)\sum_{j=1}^N \bar{c}_j(x_j^*)\sum_{j=1}^N v_j\sqrt{2\ln\frac{\pi^2 NQT^3}{3\delta'}}\sigma$$

$$+\sum_{j=1}^N c_j(x_j^*)\sum_{j=1}^N v_j\,n_j(x_j^*)-\sum_{j=1}^N \bar{c}_j(x_j^*)\sum_{j=1}^N v_j\,n_j(x_j^*)+$$

$$\frac{2n_{\max}}{\beta_{opt}^2}\sum_{j=1}^N c_j(x_j^*)\sum_{j=1}^N \bar{c}_j(x_j^*)\sum_{j=1}^N v_j\sqrt{2\ln\frac{\pi^2 NQT^3}{3\delta'}}\sigma<0\Bigg) \tag{54}$$

$$\le\mathbb{P}\Bigg(\sum_{j=1}^N v_j\,\underline{n}_j(x_j^*)-\sum_{j=1}^N v_j\,n_j(x_j^*)+2\underbrace{\frac{\sum_{j=1}^N \bar{c}_j(x_j^*)}{\beta_{opt}}}_{\ge 1}\sum_{j=1}^N v_j\sqrt{2\ln\frac{\pi^2 NQT^3}{3\delta'}}\sigma<0\Bigg)$$

$$+\mathbb{P}\Bigg(\sum_{j=1}^N c_j(x_j^*)\sum_{j=1}^N v_j\,n_j(x_j^*)-\sum_{j=1}^N \bar{c}_j(x_j^*)\sum_{j=1}^N v_j\,n_j(x_j^*)$$

$$+2\underbrace{\frac{\sum_{j=1}^N c_j(x_j^*)\sum_{j=1}^N \bar{c}_j(x_j^*)}{\beta_{opt}^2}}_{\ge 1}\sum_{j=1}^N v_j\underbrace{n_{\max}}_{\ge n_j(x_j^*)}\sqrt{2\ln\frac{\pi^2 NQT^3}{3\delta'}}\sigma<0\Bigg) \tag{55}$$

$$\le\sum_{j=1}^N\mathbb{P}\Bigg(\underline{n}_j(x_j^*)-n_j(x_j^*)+2\sqrt{2\ln\frac{\pi^2 NQT^3}{3\delta'}}\sigma\le 0\Bigg)$$

$$+\sum_{j=1}^N\mathbb{P}\Bigg(c_j(x_j^*)-\bar{c}_j(x_j^*)+2\sqrt{2\ln\frac{\pi^2 NQT^3}{3\delta'}}\sigma<0\Bigg) \tag{56}$$

$$\le\sum_{j=1}^N\mathbb{P}\Bigg(\hat{n}_{j,t-1}(x_j^*)-\sqrt{b_t}\hat{\sigma}_{j,t-1}^n(x_j^*)-n_j(x_j^*)+2\underbrace{\sqrt{2\ln\frac{\pi^2 NQT^3}{3\delta'}}\sigma}_{\ge\sqrt{b_t}\hat{\sigma}_{j,t-1}^n(x_j^*)}<0\Bigg)$$

$$+\sum_{j=1}^N\mathbb{P}\Bigg(c_j(x_j^*)-\hat{c}_{j,t-1}(x_j^*)-\sqrt{b_t}\hat{\sigma}_{j,t-1}^c(x_j^*)+2\underbrace{\sqrt{2\ln\frac{\pi^2 NQT^3}{3\delta'}}\sigma}_{\ge\sqrt{b_t}\hat{\sigma}_{j,t-1}^c(x_j^*)}<0\Bigg) \tag{57}$$

$$\le\sum_{j=1}^N\mathbb{P}\Bigg(n_j(x_j^*)<\hat{n}_{j,t-1}(x_j^*)+\sqrt{2\ln\frac{\pi^2 NQT^3}{3\delta'}}\hat{\sigma}_{j,t-1}^n(x_j^*)\Bigg)$$

$$+\sum_{j=1}^N\mathbb{P}\Bigg(c_j(x_j^*)<\hat{c}_{j,t-1}(x_j^*)-\sqrt{2\ln\frac{\pi^2 NQT^3}{3\delta'}}\hat{\sigma}_{j,t-1}^c(x_j^*)\Bigg) \tag{58}$$

$$=\sum_{j=1}^N\mathbb{P}\Bigg(\frac{n_j(x_j^*)-\hat{n}_{j,t-1}(x_j^*)}{\hat{\sigma}_{j,t-1}^n(x_j^*)}>\sqrt{2\ln\frac{\pi^2 NQT^3}{3\delta'}}\Bigg)$$

$$+ \sum_{j=1}^{N} \mathbb{P} \left( \frac{\hat{c}_{j,t-1}(x_j^*) - c_j(x_j^*)}{\hat{\sigma}_{j,t-1}^c(x_j^*)} > \sqrt{2 \ln \frac{\pi^2 NQT^3}{3\delta'}} \right) \tag{59}$$

$$\leq 2 \sum_{j=1}^{N} \frac{3\delta'}{\pi^2 NQT^3} = \frac{6\delta'}{\pi^2 QT^3}, \tag{60}$$

where in Equation (58) we used the fact that $\frac{\pi^2 NQt^2 T}{3\delta} \leq \frac{\pi^2 NQT^3}{3\delta'}$ for each $t \in \{1, \dots, T\}$, $\hat{\sigma}_{j,t-1}^n(x_j^*) \leq \sigma$ for each $j$ and $t$, and the inequality in Equation (60) is from Srinivas et al. (2010). Summing over the time horizon $T$ ensures that the optimal solution of the original problem $\{x_j^*\}_{j=1}^{N}$ is excluded from the feasible solutions at most with probability $\frac{6\delta'}{\pi^2 QT^2}$. Finally, using a union bound, we have that the optimal solution can be chosen over the time horizon with probability at least $1 - \frac{3\delta'}{\pi^2 QT^2} - \frac{6\delta'}{\pi^2 QT^2} \leq 1 - \frac{\delta'}{QT^2}$.

Notice that here we want to compute the regret of the $\mathsf{GCB}_{\mathtt{safe}}$ algorithm w.r.t. $\{x_j^*\}_{j=1}^{N}$, which is not optimal for the analysed relaxed problem. Nonetheless, the proof on the pseudo-regret provided in Theorem 4 is also valid for suboptimal solutions in the case it is feasible with high probability. This can be trivially shown using the fact that the regret w.r.t. a generic solution cannot be larger than the one computed w.r.t. the optimal one. Thanks to that, using a union bound over the probability that the bounds hold and that $\{x_j^*\}_{j=1}^{N}$ is feasible, we conclude that with probability at least $1 - \delta - \frac{\delta'}{QT^2}$ the regret $\mathsf{GCB}_{\mathtt{safe}}$ is of the order of $\mathcal{O}\left( \sqrt{T \sum_{j=1}^{N} \gamma_{j,T}} \right)$. Finally, thanks to the property of the $\mathsf{GCB}_{\mathtt{safe}}$ algorithm shown in Theorem 7, the learning policy is $\delta$-safe for the relaxed problem. $\qquad \square$

**Theorem 11** ($\mathsf{GCB}_{\mathtt{safe}}(0, \phi)$ pseudo-regret and safety with tolerance). *When:*

$$\phi \geq 2N \sqrt{2 \ln \left( \frac{\pi^2 NQT^3}{3\delta'} \right)} \sigma$$

*and*

$$\lambda_{opt} > \lambda + \frac{(\beta + n_{\max}) \phi \sum_{j=1}^{N} v_j}{N\beta^2},$$

*where $\delta' \leq \delta$, and $n_{\max} := \max_{j,x} n_j(x)$ is maximum expected number of clicks, $\mathsf{GCB}_{\mathtt{safe}}(0, \phi)$ provides a pseudo-regret w.r.t. the optimal solution to the original problem of $\mathcal{O}\left( \sqrt{T \sum_{j=1}^{N} \gamma_{j,T}} \right)$ with probability at least $1 - \delta - \frac{6\delta'}{\pi^2 QT^2}$, while being $\delta$-safe w.r.t. the constraints of the auxiliary problem.*

*Proof.* We show that at a specific day $t$ since the optimal solution of the original problem $\{x_j^*\}_{j=1}^{N}$ is included in the set of feasible ones, we are in a setting analogous to the one of $\mathsf{GCB}$, in which the regret is sublinear. Let us assume that the upper bounds to all the quantities (number of clicks and costs) holds. This has been shown before to occur with overall probability $\delta$ over the whole time horizon $T$.

First, let us evaluate the probability that the optimal solution is not feasible. This occurs if its bounds are either violating the ROI or budget constraints. From the fact that the ROI of the optimal solution satisfies $\lambda_{opt} > \lambda + \frac{(\beta + n_{\max}) \phi \sum_{j=1}^{N} v_j}{N\beta^2}$, we have:

$$\mathbb{P} \left( \frac{\sum_{j=1}^{N} v_j \, \underline{n}_j(x_j^*)}{\sum_{j=1}^{N} \overline{c}_j(x_j^*)} < \lambda \right) \tag{61}$$

$$\leq \mathbb{P} \left( \frac{\sum_{j=1}^{N} v_j \, \underline{n}_j(x_j^*)}{\sum_{j=1}^{N} \overline{c}_j(x_j^*)} < \lambda_{opt} - \frac{(\beta + n_{\max}) \phi \sum_{j=1}^{N} v_j}{N\beta^2} \right) \tag{62}$$

$$= \mathbb{P} \left( \frac{\sum_{j=1}^{N} v_j \, \underline{n}_j(x_j^*)}{\sum_{j=1}^{N} \overline{c}_j(x_j^*)} < \frac{\sum_{j=1}^{N} v_j \, n_j(x_j^*)}{\sum_{j=1}^{N} c_j(x_j^*)} - 2 \frac{\beta_{opt} + n_{\max}}{\beta_{opt}^2} \sum_{j=1}^{N} v_j \sqrt{\ln \frac{\pi^2 NQT^3}{3\delta'}} \sigma \right) \tag{63}$$

$$\leq \frac{3\delta'}{\pi^2 QT^3}, \tag{64}$$

where the derivation uses arguments similar to the ones applied in the proof for the ROI constraint in Theorem 10. Summing over the time horizon $T$ ensures that the optimal solution of the original problem $\left\{x_j^*\right\}_{j=1}^N$ is excluded from the feasible solutions at most with probability $\frac{3\delta'}{\pi^2 QT^2}$.

Second, let us evaluate the probability for which the optimal solution of the original problem $\left\{x_j^*\right\}_{j=1}^N$ is excluded due to the budget constraint, formally:

$$\mathbb{P}\left(\sum_{j=1}^N \overline{c}_j(x_j^*) > \beta + \phi\right) \tag{65}$$

$$\leq \mathbb{P}\left(\sum_{j=1}^N \overline{c}_j(x_j^*) > \beta + 2N\sqrt{2\ln\frac{\pi^2 NQT^3}{3\delta'}}\sigma\right) \tag{66}$$

$$= \mathbb{P}\left(\sum_{j=1}^N \overline{c}_j(x_j^*) > \sum_{j=1}^N c_j(x_j^*) + 2N\sqrt{2\ln\frac{\pi^2 NQT^3}{3\delta'}}\sigma\right) \tag{67}$$

$$\leq \sum_{j=1}^N \mathbb{P}\left(\overline{c}_j(x_j^*) > c_j(x_j^*) + 2\sqrt{\ln\frac{12NT^3}{\pi^2\delta'}}\sigma\right) \tag{68}$$

$$= \sum_{j=1}^N \mathbb{P}\left(\hat{c}_{j,t-1}(x_j^*) - c_j(x_j^*) \geq -\sqrt{b_t}\hat{\sigma}_{j,t-1}^c(x_j^*) + 2\sqrt{2\ln\frac{\pi^2 NQT^3}{3\delta'}}\sigma\right) \tag{69}$$

$$\leq \sum_{j=1}^N \mathbb{P}\left(\hat{c}_{j,t-1}(x_j^*) - c_j(x_j^*) \geq \sqrt{2\ln\frac{\pi^2 NQT^3}{3\delta'}}\hat{\sigma}_{j,t-1}^c(x_j^*)\right) \tag{70}$$

$$\leq \sum_{j=1}^N \mathbb{P}\left(\frac{\hat{c}_{j,t-1}(x_j^*) - c_j(x_j^*)}{\hat{\sigma}_{j,t-1}^c(x_j^*)} \geq \sqrt{2\ln\frac{\pi^2 NQT^3}{3\delta'}}\right) \tag{71}$$

$$\leq \sum_{j=1}^N \frac{3\delta'}{\pi^2 NQT^3} = \frac{3\delta'}{\pi^2 QT^3}, \tag{72}$$

where we use the fact that $\beta = \beta_{opt}$, and the derivation uses arguments similar to the ones applied in the proof for the budget constraint in Theorem 10. Summing over the time horizon $T$, we get that the optimal solution of the original problem $\left\{x_j^*\right\}_{j=1}^N$ is excluded from the set of the feasible ones with probability at most $\frac{\pi^2\delta'}{6T^2}$. Finally, using a union bound, we have that the optimal solution can be chosen over the time horizon with probability at least $1 - \frac{3\delta'}{\pi^2 QT^2}$.

Notice that here we want to compute the regret of the $\mathsf{GCB_{safe}}$ algorithm w.r.t. $\left\{x_j^*\right\}_{j=1}^N$ which is not optimal for the analysed relaxed problem. Nonetheless, the proof on the pseudo-regret provided in Theorem 4 is valid also for suboptimal solutions in the case it is feasible with high probability. This can be trivially shown using the fact that the regret w.r.t. a generic solution cannot be larger than the one computed on the optimal one. Thanks to that, using a union bound over the probability that the bounds hold and that $\left\{x_j^*\right\}_{j=1}^N$ is feasible, we conclude that with probability at least $1 - \delta - \frac{6\delta'}{\pi^2 QT^2}$ the regret $\mathsf{GCB_{safe}}$ is of the order of $\mathcal{O}\left(\sqrt{T\sum_{j=1}^N \gamma_{j,T}}\right)$. Finally, thanks to the property of the $\mathsf{GCB_{safe}}$ algorithm shown in Theorem 7, the learning policy is $\delta$-safe for the relaxed problem. $\qquad\square$

**Theorem 8** ($\mathsf{GCB_{safe}}(\psi, \phi)$ pseudo-regret and safety with tolerance). *Setting*

$$\psi = 2\frac{\beta_{opt} + n_{\max}}{\beta_{opt}^2}\sigma\sum_{j=1}^N v_j\sqrt{2\ln\left(\frac{\pi^2 NQT^3}{3\delta'}\right)} \qquad and \qquad \phi = 2N\sigma\sqrt{2\ln\left(\frac{\pi^2 NQT^3}{3\delta'}\right)},$$

*where $\delta' \leq \delta$, $\mathsf{GCB}_{\mathtt{safe}}(\psi, \phi)$ provides a pseudo-regret w.r.t. the optimal solution to the original problem of $\mathcal{O}\left(\sqrt{T \sum_{j=1}^{N} \gamma_{j,T}}\right)$ with probability at least $1 - \delta - \frac{\delta'}{QT^2}$, while being $\delta$-safe w.r.t. the constraints of the auxiliary problem.*

*Proof.* The proof follows from combining the arguments about the ROI constraint used in Theorem 10 and those about the budget constraint used in Theorem 11. □

## C    RUNNING TIME

The asymptotic running time of the $\mathsf{GCB}$ and $\mathsf{GCB}_{\mathtt{safe}}$ algorithms is given by the summation of the running time of the estimation and optimization subroutine.

The asymptotic running time of the estimation procedure, whose main component is the estimation of the quantities in Equations (2)-(5) is $\Theta(\sum_{j=1}^{N} |X_j| \, t^2)$, where $t$ is the number of samples (corresponding to the rounds), and the asymptotic space complexity is $\Theta(Nt^2)$, *i.e.*, the space required to store the Gram matrix. A better (linear) dependence on the number of days $t$ can be obtained by using the recursive formula for the GP mean and variance computation (see Chowdhury & Gopalan (2017) for details).

The asymptotic running time of the $\mathsf{Opt}$ procedure is $\Theta\left(\sum_{j=1}^{N} |X_j| \, |Y|^2 \, |R|^2\right)$, where $|X_j|$ is the cardinality of the set of bids $X_j$, since it cycles over all the subcampaigns and, for each one of them, finds the maximum bids and compute the values in the matrix $S(y, r)$. Moreover, the asymptotic space complexity of the $\mathsf{Opt}$ procedure is $\Theta\left(\max_{j=\{1,...,N\}} |X_j| \, |Y| \, |R|\right)$ since it stores the values in the matrix $S(y, r)$ and finds the maximum over the possible bids $x \in X_j$.

## D    ADDITIONAL EXPERIMENTS AND EXPERIMENTAL SETTINGS DETAILS

### D.1    EXPERIMENT #2: EVALUATING $\mathsf{GCB}_{\mathtt{safe}}(\psi, 0)$ WHEN THE BUDGET CONSTRAINT IS ACTIVE

In real-world scenarios, the business goals in terms of volumes-profitability tradeoff are often blurred, and sometimes it can be desirable to slightly violate the constraints (usually, the ROI constraint) in favor of a significant volume increase. However, analyzing and acquiring information about these tradeoff curves requires exploring volumes of opportunities by relaxing the constraints. In this experiment, we show how our approach can be adjusted to address this problem in practice.

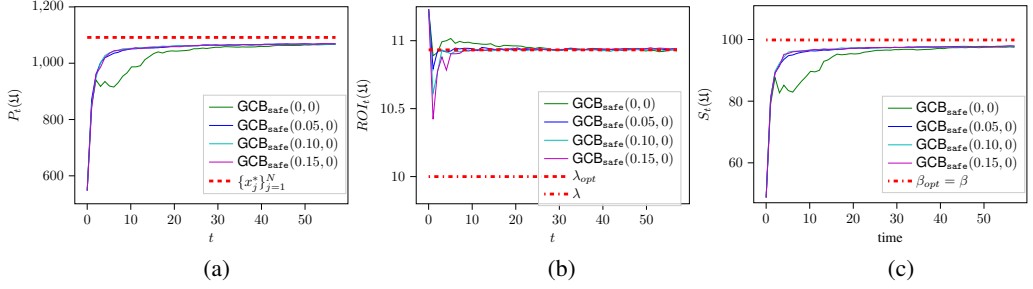

Figure 2: Results of Experiment #2: Median values of the daily revenue (a), ROI (b) and spend (c) obtained by $\mathsf{GCB}_{\mathtt{safe}}(\psi, 0)$ with different values of $\psi$.

**Setting**    We use the same setting of Experiment #1, except that we evaluate $\mathsf{GCB}_{\mathtt{safe}}$ and $\mathsf{GCB}_{\mathtt{safe}}(\psi, \phi)$ algorithms. More precisely, we relax the ROI constraint by a tolerance $\psi \in \{0, 0.05, 0.1, 0.15\}$ (while keeping $\phi = 0$). Notice that $\mathsf{GCB}_{\mathtt{safe}}(0, 0)$ corresponds to the use of $\mathsf{GCB}_{\mathtt{safe}}$ in the original problem. As a result, except for the case $\phi = 0$, we allow $\mathsf{GCB}_{\mathtt{safe}}(\psi, \phi)$ to violate the ROI constraint, but, with high probability, the violation is bounded by at most 0.5%, 1%, 1.5% of $\lambda$, respectively. Instead, we do not introduce any tolerance for the daily budget constraint $\beta$.

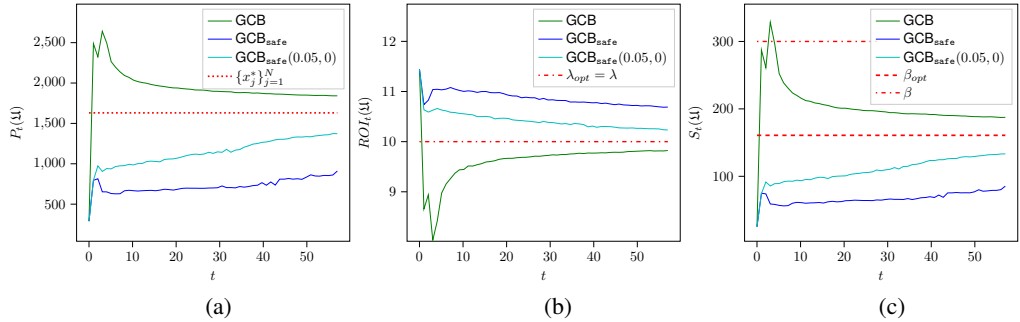

Figure 3: Results of Experiment #3: Median values of the daily revenue (a), ROI (b) and spend (c) of GCB, GCB$_\texttt{safe}$, and GCB$_\texttt{safe}$(0.05, 0).

**Results** In Figures 2, we show the median values, on 100 independent runs, of the performance in terms of daily revenue, ROI, and spend of GCB$_\texttt{safe}(\psi, 0)$ for every value of $\psi$. The 10% and 90% quantiles are reported in Figure 4, 5, 6, and 7 in Appendix D.6. The results show that by allowing a small tolerance in the ROI constraint violation, we can improve the exploration and, therefore, lead to faster convergence. We note that if we set a value of $\psi \geq 0.05$, we achieve better performance in the first learning steps ($t < 20$), still maintaining a robust behavior in terms of constraint violations. Most importantly, the ROI constraint is always satisfied by the median and also by the 10% and 90% quantiles. Furthermore, a few violations are present only in the early stages of the learning process.

## D.2 EXPERIMENT #3: COMPARING GCB, GCB$_\texttt{safe}$, AND GCB$_\texttt{safe}(\psi, 0)$ WHEN THE ROI CONSTRAINT IS ACTIVE

We study a setting in which the ROI constraint is active at the optimal solution, *i.e.*, $\lambda = \lambda_{opt}$, while the budget constraint is not. This means that, at the optimal solution, the advertiser would have an extra budget to spend. However, if such a budget is not spent, the ROI constraint would be violated otherwise.

**Setting** The experimental setting is the same as Experiment #1, except that we set the budget constraint as $\beta = 300$. The optimal daily spend is $\beta_{opt} = 161$.

**Results** In Figure 3, we show the median values of the daily revenue, the ROI, and the spend of GCB, GCB$_\texttt{safe}$, GCB$_\texttt{safe}$(0.05, 0) obtained with 100 independent runs. The 10% and 90% of the quantities provided by GCB, GCB$_\texttt{safe}$, and GCB$_\texttt{safe}$(0.05, 0) are reported in Figures 8, 9, and 10 in D.7. We notice that, even in this setting, GCB violates the ROI constraint for the entire time horizon, and the budget constraint in $t = 6$ and $t = 7$. However, it achieves a revenue larger than that of the optimal constrained solution. On the other side, GCB$_\texttt{safe}$ and always satisfies both the constraints, but it does not perform enough exploration to quickly converge to the optimal solution. We observe that it is sufficient to allow a tolerance in the ROI constraint violation by slightly perturbing the input value $\lambda$ ($\psi = 0.05$, corresponding to a violation of the constraint by at most 0.5%) to make GCB$_\texttt{safe}(\psi, \phi)$ capable of approaching the optimal solution while satisfying both constraints for every $t \in \{0, \dots, T\}$. This suggests that, in real-world applications, GCB$_\texttt{safe}(\psi, \phi)$ with a small tolerance represents an effective solution, providing guarantees on the violation of the constraints while returning high values of revenue.

## D.3 EXPERIMENT #4: COMPARING GCB, GCB$_\texttt{safe}$, AND GCB$_\texttt{safe}(\psi, \phi)$ WITH MULTIPLE, HETEROGENEOUS SETTINGS

In this experiment, we extend the experimental activity we conduct in Experiments #1 and #3 to other multiple, heterogeneous settings.

**Setting** We simulate $N = 5$ subcampaigns with a daily budget $\beta = 100$, with $|X_j| = 201$ bid values evenly spaced in $[0, \ 2]$, $|Y| = 101$ cost values evenly spaced in $[0, \ 100]$, being the daily

Table 1: Results of Experiment #4.

| | | $W_T$ | $W_{T/2}$ | $\sigma_T$ | $\sigma_{T/2}$ | $M_T$ | $M_{T/2}$ | $U_T$ | $U_{T/2}$ | $L_T$ | $L_{T/2}$ | $V_{ROI}$ | $V_B$ |
|---|---|---|---|---|---|---|---|---|---|---|---|---|---|
| Setting #1 | GCB | 57481 | 30767 | 556 | 376 | 57497 | 30811 | 58081 | 31239 | 56758 | 30288 | 1.00 | 0.62 |
| | GCB$_{\text{safe}}$ | 44419 | 21549 | 4766 | 2474 | 45348 | 21972 | 46783 | 23163 | 42287 | 20324 | 0.02 | 0.00 |
| | GCB$_{\text{safe}}(0.05,0)$ | 48028 | 23524 | 4902 | 2487 | 48626 | 23831 | 50388 | 24827 | 46307 | 22506 | 0.21 | 0.00 |
| | GCB$_{\text{safe}}(0.10,0)$ | 52327 | 25859 | 829 | 611 | 52338 | 25887 | 53324 | 26605 | 51316 | 25104 | 0.94 | 0.00 |
| Setting #2 | GCB | 63664 | 35566 | 1049 | 679 | 63701 | 35573 | 64984 | 36524 | 62249 | 34675 | 1.00 | 0.14 |
| | GCB$_{\text{safe}}$ | 34675 | 16290 | 8541 | 4448 | 37028 | 17647 | 39594 | 19473 | 27748 | 11141 | 0.03 | 0.00 |
| | GCB$_{\text{safe}}(0.05,0)$ | 40962 | 19564 | 6013 | 3122 | 41823 | 20152 | 44468 | 21698 | 38640 | 17645 | 0.04 | 0.00 |
| | GCB$_{\text{safe}}(0.10,0)$ | 46694 | 22099 | 6382 | 3112 | 47749 | 22433 | 51564 | 24776 | 44099 | 19929 | 0.72 | 0.00 |
| Setting #3 | GCB | 54845 | 30213 | 757 | 478 | 54816 | 30177 | 55734 | 30885 | 54006 | 29638 | 1.00 | 0.25 |
| | GCB$_{\text{safe}}$ | 35726 | 16577 | 8239 | 4361 | 38302 | 18114 | 40746 | 19882 | 27279 | 8791 | 0.03 | 0.00 |
| | GCB$_{\text{safe}}(0.05,0)$ | 38757 | 18370 | 8492 | 4594 | 41422 | 19808 | 43337 | 21092 | 30413 | 12678 | 0.07 | 0.00 |
| | GCB$_{\text{safe}}(0.10,0)$ | 42184 | 19993 | 9652 | 5056 | 44820 | 21574 | 47659 | 23118 | 36570 | 14450 | 0.75 | 0.00 |
| Setting #4 | GCB | 71404 | 37383 | 351 | 262 | 71399 | 37387 | 71877 | 37732 | 70930 | 37021 | 0.98 | 0.98 |
| | GCB$_{\text{safe}}$ | 29101 | 13817 | 7052 | 3646 | 30992 | 14680 | 35602 | 17256 | 20509 | 9562 | 0.00 | 0.00 |
| | GCB$_{\text{safe}}(0.05,0)$ | 39802 | 18270 | 10232 | 4955 | 38296 | 17994 | 53375 | 24962 | 25197 | 11341 | 0.01 | 0.00 |
| | GCB$_{\text{safe}}(0.10,0)$ | 51515 | 24095 | 11094 | 5639 | 56621 | 24902 | 61992 | 30020 | 35642 | 16198 | 0.56 | 0.00 |
| Setting #5 | GCB | 74638 | 39523 | 642 | 392 | 74693 | 39529 | 75405 | 40049 | 73756 | 39063 | 0.98 | 0.31 |
| | GCB$_{\text{safe}}$ | 48956 | 23230 | 6715 | 3486 | 50021 | 23838 | 53664 | 26266 | 42946 | 19287 | 0.00 | 0.00 |
| | GCB$_{\text{safe}}(0.05,0)$ | 56205 | 27003 | 2578 | 1742 | 56554 | 27211 | 58839 | 28802 | 53278 | 24987 | 0.00 | 0.00 |
| | GCB$_{\text{safe}}(0.10,0)$ | 63411 | 30207 | 5636 | 2916 | 64364 | 30665 | 66764 | 32212 | 60519 | 28260 | 0.59 | 0.00 |
| Setting #6 | GCB | 67118 | 35775 | 327 | 260 | 67130 | 35795 | 67536 | 36111 | 66726 | 35424 | 0.98 | 0.98 |
| | GCB$_{\text{safe}}$ | 14448 | 7707 | 6006 | 3065 | 15019 | 8075 | 18581 | 9800 | 6781 | 3926 | 0.02 | 0.00 |
| | GCB$_{\text{safe}}(0.05,0)$ | 14968 | 7710 | 6174 | 2974 | 15161 | 8157 | 20548 | 10351 | 7954 | 3860 | 0.02 | 0.00 |
| | GCB$_{\text{safe}}(0.10,0)$ | 34716 | 15507 | 16133 | 7280 | 37409 | 16601 | 55236 | 25366 | 9895 | 5188 | 0.19 | 0.00 |
| Setting #7 | GCB | 63038 | 35330 | 873 | 401 | 63088 | 35367 | 64226 | 35793 | 61754 | 34823 | 1.00 | 0.41 |
| | GCB$_{\text{safe}}$ | 31662 | 14806 | 5651 | 3090 | 33009 | 15570 | 35004 | 16922 | 28296 | 11338 | 0.04 | 0.00 |
| | GCB$_{\text{safe}}(0.05,0)$ | 37744 | 17606 | 4173 | 2619 | 38321 | 18161 | 41184 | 19805 | 33914 | 15276 | 0.03 | 0.00 |
| | GCB$_{\text{safe}}(0.10,0)$ | 42528 | 20046 | 7497 | 3624 | 43765 | 20683 | 47187 | 22301 | 38988 | 18314 | 0.70 | 0.00 |
| Setting #8 | GCB | 79571 | 42322 | 476 | 375 | 79581 | 42317 | 80073 | 42743 | 78969 | 41913 | 1.00 | 0.98 |
| | GCB$_{\text{safe}}$ | 48046 | 22478 | 11779 | 6000 | 52094 | 24180 | 57321 | 28024 | 30655 | 13338 | 0.02 | 0.00 |
| | GCB$_{\text{safe}}(0.05,0)$ | 58450 | 27477 | 10296 | 5605 | 61404 | 28845 | 66902 | 32883 | 41196 | 18222 | 0.02 | 0.00 |
| | GCB$_{\text{safe}}(0.10,0)$ | 68252 | 33255 | 3436 | 2417 | 68886 | 33857 | 70758 | 35377 | 65394 | 30696 | 0.07 | 0.00 |
| Setting #9 | GCB | 70280 | 37363 | 672 | 347 | 70275 | 37352 | 71123 | 37811 | 69379 | 36942 | 1.00 | 0.34 |
| | GCB$_{\text{safe}}$ | 40116 | 18895 | 5522 | 3047 | 40673 | 19357 | 43850 | 21161 | 37310 | 17222 | 0.03 | 0.00 |
| | GCB$_{\text{safe}}(0.05,0)$ | 51138 | 23683 | 3110 | 2036 | 50984 | 23375 | 54545 | 26174 | 47465 | 21385 | 0.03 | 0.00 |
| | GCB$_{\text{safe}}(0.10,0)$ | 63574 | 29675 | 3810 | 3323 | 64011 | 30112 | 66658 | 32559 | 60970 | 27280 | 0.80 | 0.00 |
| Setting #10 | GCB | 80570 | 41973 | 435 | 344 | 80568 | 42019 | 81127 | 42388 | 80023 | 41496 | 1.00 | 0.98 |
| | GCB$_{\text{safe}}$ | 58965 | 28785 | 3097 | 1465 | 60033 | 28917 | 62353 | 30535 | 54590 | 26931 | 0.02 | 0.00 |
| | GCB$_{\text{safe}}(0.05,0)$ | 63685 | 31004 | 3787 | 1876 | 65273 | 31550 | 67364 | 33105 | 57860 | 28349 | 0.02 | 0.00 |
| | GCB$_{\text{safe}}(0.10,0)$ | 68480 | 33358 | 4224 | 2181 | 70388 | 33998 | 72730 | 35838 | 61971 | 30317 | 0.65 | 0.00 |

budget $\beta = 100$, and $|R|$ evenly spaced revenue values depending on the setting. We generate 10 scenarios that differ in the parameters defining the cost and revenue functions and in the ROI parameter $\lambda$. Recall that the number-of-click functions coincide with the revenue functions since $v_j = 1$ for each $j \in \{1, \ldots, N\}$. Parameters $\alpha_j \in \mathbb{N}^+$ and $\theta_j \in \mathbb{N}^+$ are sampled from discrete uniform distributions $\mathcal{U}\{50, 100\}$ and $\mathcal{U}\{400, 700\}$, respectively. Parameters $\gamma_j$ and $\delta_j$ are sampled from the continuous uniform distributions $\mathcal{U}[0.2, 1.1)$. Finally, parameters $\lambda$ are chosen such that the ROI constraint is active at the optimal solution. Table 3 in D.8 specifies the values of such parameters.

**Results**   We compare the GCB, GCB$_{\text{safe}}$, GCB$_{\text{safe}}(0.05, 0)$, and GCB$_{\text{safe}}(0.10, 0)$ algorithms in terms of:

- $W_t := \sum_{h=1}^{t} P_t(\mathfrak{U})$: average (over 100 runs) cumulative revenue at round $t$ (and the corresponding standard deviation $\sigma_t$);

- $M_t$: median (over 100 runs) of the cumulative revenue at round $t$;

- $U_t$: 90-th percentile (over 100 runs) of the cumulative revenue at round $t$;

- $L_t$: 10-th percentile (over 100 runs) of the cumulative revenue at round $t$;

- $V_{ROI}$: the fraction of days in which the ROI constraint is violated;

- $V_B$: the fraction of days in which the budget constraint is violated.

Table 2: Parameters of the synthetic settings used in Experiment #1.

|  | $C_1$ | $C_2$ | $C_3$ | $C_4$ | $C_5$ |
|---|---|---|---|---|---|
| $\theta_j$ | 60 | 77 | 75 | 65 | 70 |
| $\delta_j$ | 0.41 | 0.48 | 0.43 | 0.47 | 0.40 |
| $\alpha_j$ | 497 | 565 | 573 | 503 | 536 |
| $\gamma_j$ | 0.65 | 0.62 | 0.67 | 0.68 | 0.69 |
| $\sigma_f$ GP revenue | 0.669 | 0.499 | 0.761 | 0.619 | 0.582 |
| $l$ GP revenue | 0.425 | 0.469 | 0.471 | 0.483 | 0.386 |
| $\sigma_f$ GP cost | 0.311 | 0.443 | 0.316 | 0.349 | 0.418 |
| $l$ GP cost | 0.76 | 0.719 | 0.562 | 0.722 | 0.727 |

Table 1 reports the algorithms' performances at $\lceil T/2 \rceil = 28$ and at the end of the learning process $t = T = 57$. As already observed in the previous experiments, GCB violates the ROI constraint at every round, run, and setting. More surprisingly, GCB violates the budget constraint most of the time (60% on average) even if that constraint is not active at the optimal solution. Interestingly, $\mathsf{GCB}_{\mathsf{safe}}(\psi, 0)$ never violates the budget constraints (for every $\psi$). As expected, the violation of the ROI constraint is close to zero with $\mathsf{GCB}_{\mathsf{safe}}$, while it increases as $\psi$ increases. In terms of average cumulative revenue, at $T$, we observe that $\mathsf{GCB}_{\mathsf{safe}}$ gets about 56% of the revenue provided by GCB, while the ratio related to $\mathsf{GCB}_{\mathsf{safe}}(0.05, 0)$ is about 66% and that related to $\mathsf{GCB}_{\mathsf{safe}}(0.10, 0)$ is about 78%. At $T/2$, we the ratios are about 52% for GCB, 61% for $\mathsf{GCB}_{\mathsf{safe}}(0.05, 0)$, and 73% for $\mathsf{GCB}_{\mathsf{safe}}(0.10, 0)$, showing that those ratios increase as $T$ increases. The rationale is that in the early stages of the learning process, safe algorithms learn more slowly than non-safe algorithms. Similar performances can be observed when focusing on the other indices. Summarily, the above results show that our algorithms provide advertisers with a wide spectrum of effective tools to address the revenue/safety tradeoff. A small value of $\psi$ (and $\phi$) represents a good tradeoff. The choice of the specific configuration to adopt in practice depends on the advertiser's aversion to the violation of the constraints.

### D.4 ADDITIONAL INFORMATION FOR REPRODUCIBILITY

In this section, we provide additional information for the full reproducibility of the experiments provided in the main paper.

The code has been run on an Intel(R) Core(TM) $i7 - 4710MQ$ CPU with 16 GiB of system memory. The operating system was Ubuntu 18.04.5 LTS, and the experiments have been run on Python 3.7.6. The libraries used in the experiments, with the corresponding versions, were:

- `matplotlib==3.1.3`
- `gpflow==2.0.5`
- `tikzplotlib==0.9.4`
- `tf_nightly==2.2.0.dev20200308`
- `numpy==1.18.1`
- `tensorflow_probability==0.10.0`
- `scikit_learn==0.23.2`
- `tensorflow==2.3.0`

On this architecture, the average execution time of each algorithm takes an average of $\approx 30$ sec for each day $t$ of execution.

### D.5 PARAMETERS AND SETTING OF EXPERIMENT #1 (MAIN PAPER)

Table 2 specifies the values of the parameters of cost and number-of-click functions of the subcampaigns used in Experiment #1.

### D.6 ADDITIONAL RESULTS OF EXPERIMENT #2

In Figures 4, 5, 6, and 7 we report the $90\%$ and $10\%$ of the quantities related to Experiment #2 provided by the $\mathsf{GCB_{safe}}$, $\mathsf{GCB_{safe}}(0, 0.05)$, $\mathsf{GCB_{safe}}(0, 0.10)$, and $\mathsf{GCB_{safe}}(0, 0.15)$, respectively.

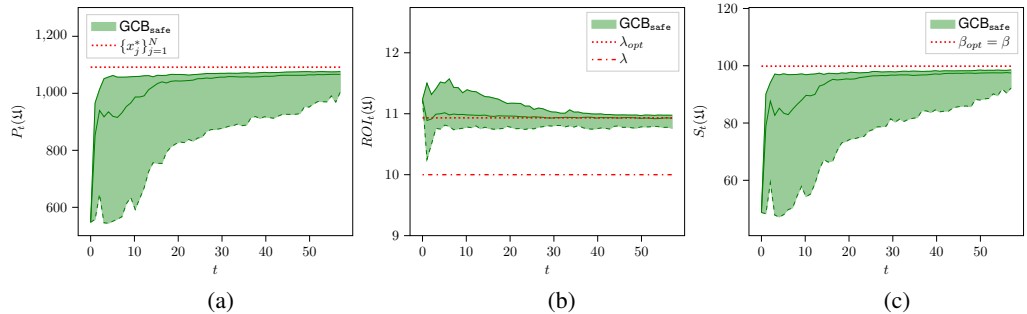

Figure 4: Results of Experiment #2: daily revenue (a), ROI (b), and spend (c) obtained by $\mathsf{GCB_{safe}}$. The dash-dotted lines correspond to the optimum values for the revenue and ROI, while the dashed lines correspond to the values of the ROI and budget constraints.

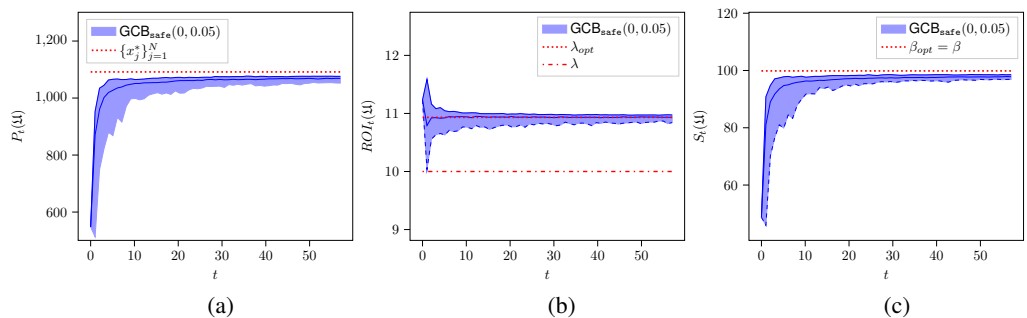

Figure 5: Results of Experiment #2: daily revenue (a), ROI (b), and spend (c) obtained by and $\mathsf{GCB_{safe}}(0, 0.05)$. The dash-dotted lines correspond to the optimum values for the revenue and ROI, while the dashed lines correspond to the values of the ROI and budget constraints.

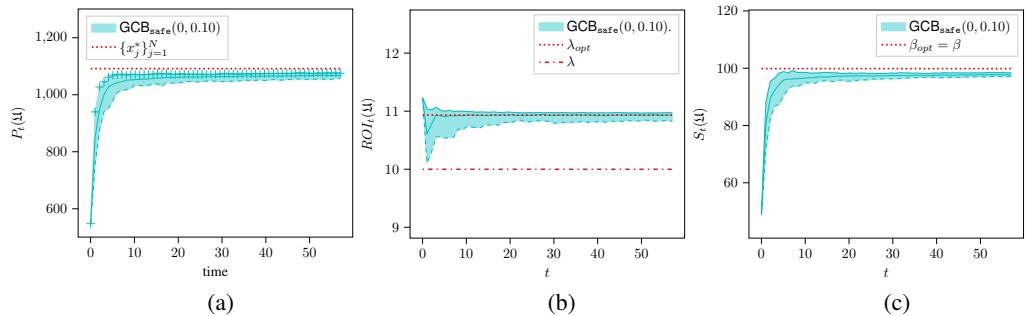

Figure 6: Results of Experiment #2: daily revenue (a), ROI (b), and spend (c) obtained by and $\mathsf{GCB_{safe}}(0, 0.10)$. The dash-dotted lines correspond to the optimum values for the revenue and ROI, while the dashed lines correspond to the values of the ROI and budget constraints.

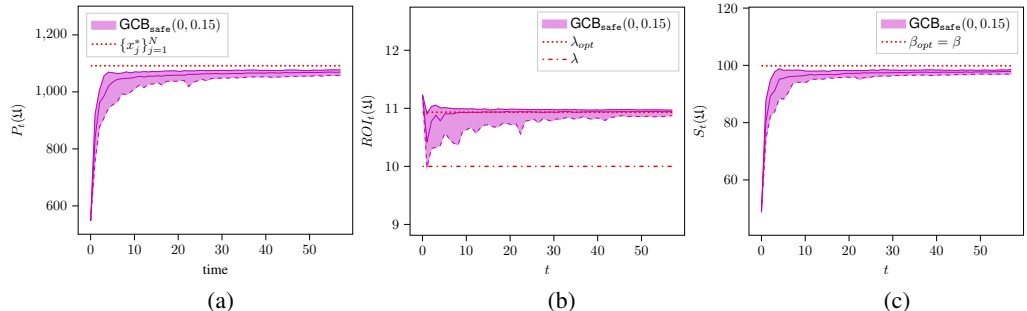

(a)  (b)  (c)

Figure 7: Results of Experiment #2: daily revenue (a), ROI (b), and spend (c) obtained by and $\text{GCB}_{\texttt{safe}}(0, 0.15)$. The dash-dotted lines correspond to the optimum values for the revenue and ROI, while the dashed lines correspond to the values of the ROI and budget constraints.

### D.7  ADDITIONAL RESULTS OF EXPERIMENT #3

In Figures 8, 9, and 10 we report the $90\%$ and $10\%$ of the quantities analysed in the experimental section for Experiment #3 provided by the GCB, $\text{GCB}_{\texttt{safe}}$, and $\text{GCB}_{\texttt{safe}}(0.05, 0)$, respectively. These results show that the constraints are satisfied by $\text{GCB}_{\texttt{safe}}$, and $\text{GCB}_{\texttt{safe}}(0.05, 0)$ also with high probability. While for $\text{GCB}_{\texttt{safe}}$ this is expected due to the theoretical results we provided, the fact that also $\text{GCB}_{\texttt{safe}}(0.05, 0)$ guarantees safety w.r.t. the original optimization problem suggests that in some specific setting $\text{GCB}_{\texttt{safe}}$ is too conservative. This is reflected in a lower cumulative revenue, which might be negative from a business point of view.

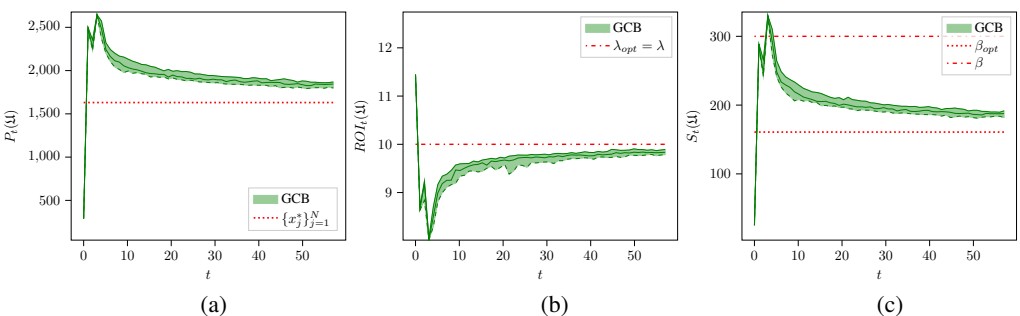

(a)  (b)  (c)

Figure 8: Results of Experiment #3: daily revenue (a), ROI (b), and spend (c) obtained by GCB. The dash-dotted lines correspond to the optimum values for the revenue and ROI, while the dashed lines correspond to the values of the ROI and budget constraints.

### D.8  PARAMETERS OF SETTINGS OF EXPERIMENT #4

We report in Table 3 the values of the parameters of cost and number-of-click functions of the subcampaigns used in Experiment #4.

Table 3: Values of the parameters used in the 10 different settings of Experiment #4.

| | | $C_1$ | $C_2$ | $C_3$ | $C_4$ | $C_5$ | $\lambda$ |
|---|---|---|---|---|---|---|---|
| Setting 1 | $\theta_j$ | 530 | 417 | 548 | 571 | 550 | 10.0 |
| | $\delta_j$ | 0.356 | 0.689 | 0.299 | 0.570 | 0.245 | |
| | $\alpha_j$ | 83 | 97 | 72 | 100 | 96 | |
| | $\gamma_j$ | 0.939 | 0.856 | 0.484 | 0.661 | 0.246 | |
| Setting 2 | $\theta_j$ | 597 | 682 | 698 | 456 | 444 | 14.0 |
| | $\delta_j$ | 0.202 | 0.520 | 0.367 | 0.393 | 0.689 | |
| | $\alpha_j$ | 83 | 98 | 56 | 60 | 51 | |
| | $\gamma_j$ | 0.224 | 0.849 | 0.726 | 0.559 | 0.783 | |
| Setting 3 | $\theta_j$ | 570 | 514 | 426 | 469 | 548 | 10.5 |
| | $\delta_j$ | 0.217 | 0.638 | 0.694 | 0.391 | 0.345 | |
| | $\alpha_j$ | 97 | 78 | 53 | 80 | 82 | |
| | $\gamma_j$ | 0.225 | 0.680 | 1.051 | 0.412 | 0.918 | |
| Setting 4 | $\theta_j$ | 487 | 494 | 467 | 684 | 494 | 12.0 |
| | $\delta_j$ | 0.348 | 0.424 | 0.326 | 0.722 | 0.265 | |
| | $\alpha_j$ | 62 | 79 | 76 | 69 | 99 | |
| | $\gamma_j$ | 0.460 | 1.021 | 0.515 | 0.894 | 1.056 | |
| Setting 5 | $\theta_j$ | 525 | 643 | 455 | 440 | 600 | 14.0 |
| | $\delta_j$ | 0.258 | 0.607 | 0.390 | 0.740 | 0.388 | |
| | $\alpha_j$ | 52 | 87 | 68 | 99 | 94 | |
| | $\gamma_j$ | 0.723 | 0.834 | 1.054 | 1.071 | 0.943 | |
| Setting 6 | $\theta_j$ | 617 | 518 | 547 | 567 | 576 | 11.0 |
| | $\delta_j$ | 0.844 | 0.677 | 0.866 | 0.252 | 0.247 | |
| | $\alpha_j$ | 71 | 53 | 87 | 98 | 59 | |
| | $\gamma_j$ | 0.875 | 0.841 | 1.070 | 0.631 | 0.288 | |
| Setting 7 | $\theta_j$ | 409 | 592 | 628 | 613 | 513 | 11.5 |
| | $\delta_j$ | 0.507 | 0.230 | 0.571 | 0.359 | 0.307 | |
| | $\alpha_j$ | 77 | 78 | 91 | 50 | 71 | |
| | $\gamma_j$ | 0.810 | 0.246 | 0.774 | 0.516 | 0.379 | |
| Setting 8 | $\theta_j$ | 602 | 605 | 618 | 505 | 588 | 13.0 |
| | $\delta_j$ | 0.326 | 0.265 | 0.201 | 0.219 | 0.291 | |
| | $\alpha_j$ | 67 | 80 | 99 | 77 | 99 | |
| | $\gamma_j$ | 0.671 | 0.775 | 0.440 | 0.310 | 0.405 | |
| Setting 9 | $\theta_j$ | 486 | 684 | 547 | 419 | 453 | 13.0 |
| | $\delta_j$ | 0.418 | 0.330 | 0.529 | 0.729 | 0.679 | |
| | $\alpha_j$ | 53 | 82 | 58 | 96 | 100 | |
| | $\gamma_j$ | 0.618 | 0.863 | 0.669 | 0.866 | 0.831 | |
| Setting 10 | $\theta_j$ | 617 | 520 | 422 | 559 | 457 | 14.0 |
| | $\delta_j$ | 0.205 | 0.539 | 0.217 | 0.490 | 0.224 | |
| | $\alpha_j$ | 51 | 86 | 93 | 61 | 84 | |
| | $\gamma_j$ | 1.0493 | 0.779 | 0.233 | 0.578 | 0.562 | |

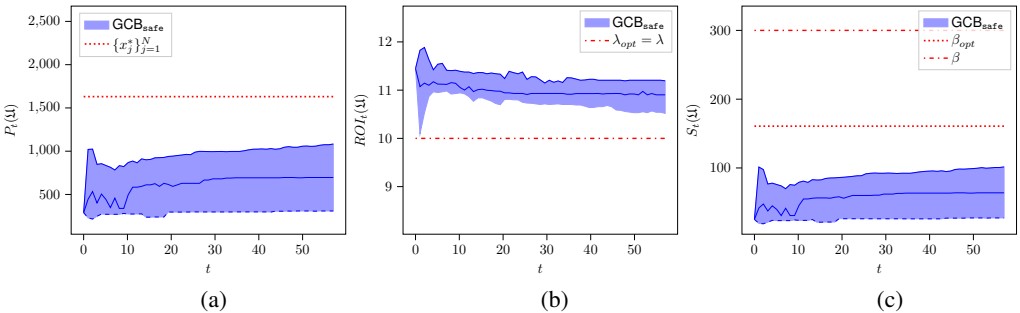

Figure 9: Results of Experiment #3: daily revenue (a), ROI (b), and spend (c) obtained by $\mathsf{GCB}_{\mathtt{safe}}$. The dash-dotted lines correspond to the optimum values for the revenue and ROI, while the dashed lines correspond to the values of the ROI and budget constraints.

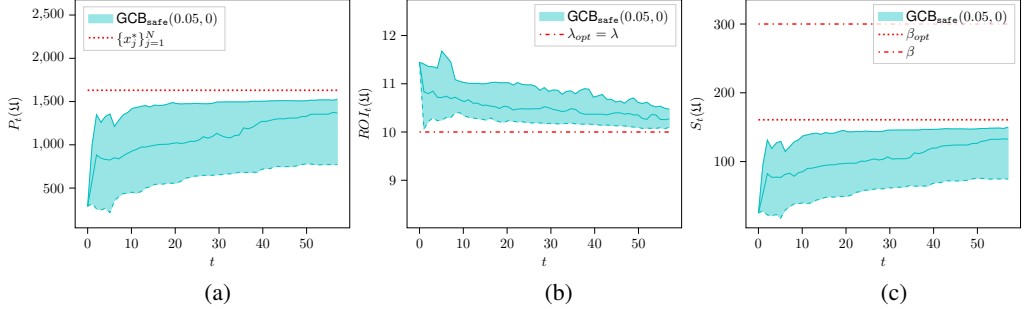

Figure 10: Results of Experiment #3: daily revenue (a), ROI (b), and spend (c) obtained by $\mathsf{GCB}_{\mathtt{safe}}(0.05, 0)$. The dash-dotted lines correspond to the optimum values for the revenue and ROI, while the dashed lines correspond to the values of the ROI and budget constraints.

