# OpenReview forum: "Safe Online Bid Optimization with Return On Investment and Budget Constraints"
_ICLR.cc/2024/Conference — Submitted to ICLR 2024_

### Official Review · Reviewer_zUb2 · 2023-10-30

**Soundness:** 3 good
**Presentation:** 3 good
**Contribution:** 2 fair
**Rating:** 3
**Confidence:** 3

**Summary:**

In this paper, the authors study a constrained optimization problem for ads campaign that maximizes the value received by a campaign subject to ROI and budget constraints. The authors first show that in an offline setting no approximation within any positive factor is possible. They then move onto an online setting where the expected number of clicks generated by a bid and expected cost given a bid becomes unknown, and propose algorithms to resolve the learning problem. In particular, the GCB algorithm guarantees sublinear regret at the cost of a linear number of constraint violations, while GCBsafe ensures constant upper bound on number of violations and a linear regret. They also show that if one is to solve a relaxed version of the constrained optimization problem, both sublinear regret and sublinear number of constraint violations is achievable. Finally, they evaluated their algorithms via numerical studies.

**Strengths:**

- The paper is well organized and written, providing both theoretical results and numerical evaluations.
- The problem that the paper considers is interesting and bears real-world relevance.
- The theoretical results provided in this paper appear sound.

**Weaknesses:**

- Regarding the constrained optimization problem considered in this paper, I wonder if the authors can provide more motivations for the forms of n_j(x) and c_j(x). Specifically, in real-world advertising these are usually determined by the mechanism of the ad auctions. Could you provide more justification for the current formulation?
- Another major concern I had is about how the authors measure the performance of the algorithm. In particular, the authors consider the number of constraint violations, but to me, a more natural and reasonable metric would be the **amount** of constraint violations for each of the campaigns. Having all of the constraints violated but only by a small amount can be better than having some of the constraints violated by a tremendous amount. I wonder if the authors have theoretical results that characterize the amount of constraint violations here.
- In Theorem 8, the authors proposed an algorithm that achieves both sublinear regret and sublinear number of constraint violations. However, as the authors discussed, the parameters $\phi$ and $\psi$ can be really large in practice. It is also unlikely that one can let the budget spent go to infinity in practice. When $\phi$ and $\psi$ become really large, the constraints essentially become meaningless. I wonder if the authors can provide more discussions for this and/or have evaluated this numerically.
- Could you also justify the formulas of n_j(x) and c_j(x) used in the numerical studies? I also wonder if the algorithms have been evaluated on real-world data.
- The related work section is not included in the paper.

**Questions:**

See weaknesses.

---

### Official Review · Reviewer_7DCz · 2023-10-30

**Soundness:** 3 good
**Presentation:** 2 fair
**Contribution:** 2 fair
**Rating:** 5
**Confidence:** 2

**Summary:**

In online marketing, advertisers face the intricate challenge of striking a delicate balance between achieving high volumes and maximizing profitability. The general objective is to optimize revenue while adhering to a minimum Return On Investment (ROI) level. This paper focuses on the intricacies of this multifaceted problem, which can be naturally captured by a combinatorial optimization problem subject to ROI and budget constraints.
The above problem is further complicated by the presence of uncertainty in the parameters of these constraints. The algorithms employed for bid allocation and strategy optimization must tackle this uncertainty, and their exploration choices can inadvertently lead to constraint violations, which act as substantial roadblocks to the effective implementation of online marketing strategies in practical, real-world scenarios.
To address these challenges, this paper conducts a comprehensive study, researching both the optimization and learning aspects of the problem. Notably, to provide practical solutions, the paper introduces novel algorithms such as GCB, GCB_safe etc., each tailored to balance regret and safety. Empirical experiments provide insights into the comparative performance of these algorithms.

**Strengths:**

The key novelty of this paper lies in introducing ROI constraints to online advertising optimization and developing multiple solution techniques which can work through learing with unknown parameters. In addition, safety is also taken into consideration for risk-sensitive applications. Specifically, several safety-centric algorithms (like GCB_safe, and GCB_safe($\varphi$,$\phi$)) are provided to balance pseudo-regret and constraint violations in uncertain environments.

For proposed algorithms, both theoretical and numerical results are provided and look reasonable to me.

**Weaknesses:**

The shortcomings of this paper mainly lie in the following aspects:

1.	ROI constraints should be better motivated by convencing examples, and I expect to see more evidences showing the importance of the problem researched in this paper. Can ROI be tackled in a different way, for example, adding a penalty factor related to ROI into the objective to be optimized? Why the way that ROI is handled in this paper is the best?
2.	Although the GCB_safe algorithm is designed not to violate ROI and budget constraints, the paper may not fully demonstrate the performance of the algorithm under variable or extreme conditions. The experimental part may lack sufficient applicaton scenario to fully evaluate the robustness and adaptability of the algorithm.

**Questions:**

see weakness

---

### Official Review · Reviewer_89GL · 2023-10-31

**Soundness:** 3 good
**Presentation:** 2 fair
**Contribution:** 2 fair
**Rating:** 3
**Confidence:** 3

**Summary:**

This paper studies the online bid learning problem in the setting of autobidding, where the goal is to maximize the (weighted) number of clicks while subject to the budget constraint and the ROI constraint. The optimization problem is formulated from a single-advertiser’s perspective and is modeled as follows:

* The advertiser has N sub-campaigns.
* For each of the sub-campaigns, the advertiser can choose one bid out of a finite set of options. Each option results in a pair of click and payment, both are stochastic.
* Across all the sub-campaigns, the total payment should not exceed the budget and the total (weighted) number of clicks over the total spend should be no less than the ROI threshold.

Negative results:
* SUBSET-SUM can be reduced to this problem, so one should not expect any efficient constraint approximation unless P = NP.
* Either the regret is at least linear in number of rounds or the constraints will be violated at least a linear number of times.

Positive results:
* An algorithm guaranteeing sublinear regret that may violate the constraints a linear number of times.
* An algorithm that violates the constraint at most a sublinear of times with linear regret bound.
* An algorithm that achieves both sublinear in regret and number of violation times while relaxing the constraints by a small margin.

Experiments with synthetic data are conducted to evaluate the proposed algorithms.

**Strengths:**

* Online bidding is an important and practical problem
* The theoretical results seem decent

**Weaknesses:**

* Insufficient literature review on highly related works (for example, [1-4])
* There are many prior works that can achieve sublinear regret and with very few or even no violations of the constraints. Although the more well-known ones are for the budget constraint only, some recent work shows that the same is achievable for the setting with both budget and ROI constraints.
* The above fact then leads to the following key questions to this work:
  * Which modeling assumptions lead to the gap? I.e., impossibility results in this work vs the positive results in prior works.
  * In which applications the modeling assumptions of this work make more sense than those of prior works?
  * What is the performance of the algorithm proposed in this work when the modeling assumptions of the prior works are applied?
* This work does not mention any work following the approach of [1-4], and does not have any discussion regarding the above questions. Which makes it very hard to evaluate the contribution.

[1] Balseiro, Santiago R., and Yonatan Gur. "Learning in repeated auctions with budgets: Regret minimization and equilibrium." Management Science 65, no. 9 (2019): 3952-3968.

[2] Balseiro, Santiago R., Haihao Lu, and Vahab Mirrokni. "The best of many worlds: Dual mirror descent for online allocation problems." Operations Research 71, no. 1 (2023): 101-119.

[3] Lucier, Brendan, Sarath Pattathil, Aleksandrs Slivkins, and Mengxiao Zhang. "Autobidders with budget and roi constraints: Efficiency, regret, and pacing dynamics." arXiv preprint arXiv:2301.13306 (2023).

[4] Balseiro, Santiago R., Kshipra Bhawalkar, Zhe Feng, Haihao Lu, Vahab Mirrokni, Balasubramanian Sivan, and Di Wang. "Joint Feedback Loop for Spend and Return-On-Spend Constraints." arXiv preprint arXiv:2302.08530 (2023).

**Questions:**

* Which modeling assumptions lead to the gap? I.e., impossibility results in this work vs the positive results in prior works.
* In which applications the modeling assumptions of this work make more sense than those of prior works?
* What is the performance of the algorithm proposed in this work when the modeling assumptions of the prior works are applied?

---

### Official Review · Reviewer_M2xs · 2023-11-02

**Soundness:** 3 good
**Presentation:** 3 good
**Contribution:** 2 fair
**Rating:** 6
**Confidence:** 3

**Summary:**

This paper investigates the online bid optimization problem under ROI and budget constraints. The authors prove some non-trivial impossibility results such as the NP-hardness of the optimization and that a sublinear regret is not achievable if the ROI or budget constraints are only violated a sublinear number of times. Then the authors propose algorithms to guarantee either a sublinear regret or a sublinear number of constraint violations. Numerical simulations are conducted to demonstrate the effectiveness of the proposed algorithms.

**Strengths:**

1. The impossibility result that no online algorithms can achieve both a sublinear number of constraint violations and a sublinear regret is strong.

2. The two MAB algorithms GCB and GCB_safe have advantages in low regret bound and small number of constraint violations, respectively.

**Weaknesses:**

1. In this paper, users' clicks are modeled as a Gaussian Process. Do we have any empirical evidence to support this assumption? Moreover, how does this assumption affect the algorithm design and regret analysis?

2. It seems to me that this work is an extension of [7]. What are the major new technical challenges of the problem studied in this paper compared to [7]? In addition, how significantly does the ROI constraint affect the difficulty of the bid optimization problem? Without clear illustrations of these questions, the technical novelty of this paper may be unclear.

3. Theorem 1 seems to be a little straightforward. The authors may want to only preserve the statement of Theorem 1 but remove its proof (or replace it with a simpler proof sketch) as it is not very interesting.

4. The experiments are numerical simulations, although I understand that this paper focuses more on theoretical analysis. If real datasets can be used in experiments, the quality of this paper can be enhanced a lot.

**Questions:**

Weakness 1 and 2

---

### Meta-Review · Area_Chair_khQj · 2023-12-04

**Metareview:**

This paper looks at some interesting online learning problem, but the reviewers were not especially thrilled.

the authors decided to skip the rebuttal, hence I take this as some implicit withdrawal

**Justification For Why Not Higher Score:**

No rebuttal or discussion, even against rejection reviews.

**Justification For Why Not Lower Score:**

N/A

---

### Decision · Program_Chairs · 2024-01-16

Reject